# Distinguishing individual photobodies using Oligopaints reveals thermo-sensitive and -insensitive phytochrome B condensation at distinct subnuclear locations

Juan Du [1], Keunhwa Kim[1,2] & Meng Chen [1] ✉

Photobodies (PBs) are membraneless subnuclear organelles that self-assemble via concentration-dependent liquid-liquid phase separation (LLPS) of the plant photoreceptor and thermosensor phytochrome B (PHYB). The current PHYB LLPS model posits that PHYB phase separates randomly in the nucleoplasm regardless of the cellular or nuclear context. Here, we established a robust Oligopaints method in *Arabidopsis* to determine the positioning of individual PBs. We show surprisingly that even in PHYB overexpression lines – where PHYB condensation would be more likely to occur randomly – PBs positioned at twelve distinct subnuclear locations distinguishable by chromocenter and nucleolus landmarks, suggesting that PHYB condensation occurs nonrandomly at preferred seeding sites. Intriguingly, warm temperatures reduce PB number by inducing the disappearance of specific thermo-sensitive PBs, demonstrating that individual PBs possess different thermosensitivities. These results reveal a nonrandom PB nucleation model, which provides the framework for the biogenesis of spatially distinct individual PBs with diverse environmental sensitivities within a single plant nucleus.

The dynamic spatial organization of the cell nucleus is a universal feature of all eukaryotes. In both animals and plants, interphase chromosomes are spatially organized at multiple scales, from the segregation of distinct chromosomal territories[1,2] to the separation of active and repressive chromatin compartments[3,4] and the recruitment of individual genes to preferred active or repressive subnuclear domains[5,6]. In addition to the nonrandom organization of chromatin, the interchromatin space harbors a variety of functionally distinct subnuclear membraneless organelles named nuclear bodies[7,8]. Extensive studies on animal nuclear bodies, including nucleoli, Cajal bodies, nuclear speckles, paraspeckles, histone locus bodies, and promyelocytic leukemia protein (PML) bodies, revealed diverse roles of nuclear bodies in transcription, splicing, and DNA replication and repair[7,8]. Nuclear bodies behave like liquid or gel droplets that are commonly considered as biomolecular condensates excluded from the surrounding environment via concentration-dependent, energetically favorable liquid-liquid phase separation (LLPS)[9,10]. Like membrane-bound organelles in the cytoplasm, nuclear bodies create spatially confined subnuclear compartments that can either selectively concentrate enzymes and reactants to enhance the specificity and efficiency of a biological reaction within the nuclear body compartment or segregate components away from their biological activities in the surrounding nucleoplasm. Unlike membrane-bound organelles, however, developmental and environmental cues can dynamically regulate the assembly and dissolution of nuclear bodies and their associated functions. However, the regulatory mechanism of LLPS in live cells remains poorly understood. Photobodies (PBs) are plant nuclear bodies defined molecularly by the presence of the photoreceptor and

¹Department of Botany and Plant Sciences, Institute for Integrative Genome Biology, University of California, Riverside, CA 92521, USA. ²Present address: Plant Molecular Biology and Biotechnology Research Center, Gyeongsang National University, Jinju 52828, Republic of Korea. ✉e-mail: meng.chen@ucr.edu

thermosensor phytochrome B (PHYB)[11,12]. Environmental light and temperature cues elicit conformational changes in PHYB to control PB formation and maintenance[12–14]. As such, the PB provides a unique opportunity to investigate the dynamic regulation of nuclear body formation and dissolution by cell signaling.

The plant genome is exquisitely sensitive to and can be intricately reprogrammed by environmental light and temperature cues to profoundly influence all aspects of plant development, growth, physiology, metabolism, and immunity, called photomorphogenesis and thermomorphogenesis, respectively[15,16]. PHYB is a central photoreceptor and thermosensor that integrates these environmental cues. PHYB is a homodimeric bilin-containing protein; each monomer comprises an N-terminal photosensory module and a C-terminal output module[17,18]. The absorption of red (R, 600 nm to 700 nm) or far-red (FR, 700 nm to 750 nm) light by a covalently linked phytochromobilin in the photosensory module photoconverts PHYB between two relatively stable conformers: an inactive, R-light-absorbing Pr form and an active, FR-light-absorbing Pfr form[17,18]. In addition to photoconversion, the active Pfr can spontaneously revert to the inactive Pr in a light-independent process called dark or thermal reversion[19]. The thermal reversion rate of PHYB can be greatly enhanced by temperature increases between 10 °C and 30 °C, which makes PHYB a thermosensor[20,21]. Therefore, both light and temperature cues directly control the Pfr/Pr equilibrium or the abundance of active PHYB. Photoactivated PHYB accumulates primarily in the nucleus to orchestrate the expression of hundreds of light- and temperature-responsive genes by regulating the stability and activity of transcription factors[11,22,23]. In particular, PHYB interacts directly with a small family of basic helix-loop-helix transcription factors called PHYTOCHROME INTERACTING FACTORs (PIFs), which are nodal regulators of plant development and growth[22,23]. As such, PHYB, through its photo and thermal conversions, quantitatively links changes in the light and temperature environment to dynamic transcriptomic programming and thereby enables environmentally-induced phenotypic plasticity in plants. PHYB signaling is best studied during seedling establishment in the plant reference species *Arabidopsis thaliana* (*Arabidopsis*). The current model posits that light and temperature changes are perceived in the epidermal cells of the embryonic leaves (cotyledons) by PHYB, which controls the biosynthesis of the mobile growth hormone auxin to regulate seedling morphogenesis, including cotyledon expansion locally, as well as the elongation of the embryonic stem (hypocotyl) remotely[24–26].

The term "photobody" was coined by Joanne Chory in reference to the dynamic assembly and dissolution of the PHYB-containing subnuclear organelles in response to R and FR light[11]. The regulation of PBs by R/FR light and temperature has been extensively studied in *Arabidopsis* pavement epidermal cells using fluorescent-protein-tagged PHYB (PHYB-FP). Akira Nagatani, Ferenc Nagy, and Eberhard Schäfer first reported the compartmentalization of photoactivated green-fluorescent-protein-tagged PHYB (PHYB-GFP) into discrete subnuclear speckles (PBs)[27,28]. These observations suggest that the condensation of the active form of PHYB drives PB formation. Consistently, PB formation and maintenance – i.e., the number and size of PHYB-FP-containing PBs – can be directly modulated by altering the abundance of active PHYB-FP through changes in the intensity of R light[29,30], the R-to-FR light ratio[29–33], or temperature[20,31]. For example, under intense R light (i.e., 10 μmol m$^{-2}$ s$^{-1}$), where each PHYB molecule stays in the Pfr at least 50 % of the time, PHYB-FP assembles into two to ten large PBs of 0.7–2 μm in diameter[29,31,32,34]. In contrast, under dim R light, where PHYB stays as the Pr for the majority of the time, PHYB-FP localizes to tens of small PBs of less than 0.1–0.7 μm in diameter, or it is dispersed evenly in the nucleoplasm[29,31,32]. PB formation relies on the dimerization of PHYB's C-terminal module[35,36]. A D1040V mutation, which disrupts PHYB dimerization, abolishes PHYB's ability to phase-separate into PBs[37]. Together, the current data support the model that

the condensation of the active dimeric form of PHYB drives PB formation and maintenance.

PB formation correlates with PHYB responses and has been considered necessary in PHYB signaling[12–14]. PBs were proposed to participate in the PHYB-mediated degradation of PIF1 and PIF3[34,38–40], sequestration of PIF7[41,42], and transcriptional regulation by TANDEM ZINC-FINGER/PLUS (TZP)[43,44]. Further supporting the functional role of PBs in PHYB signaling, recent proteomics studies have revealed that many PHYB signaling components reside in PBs, particularly those that interact directly or indirectly with PHYB[45,46]. Our recent studies suggest that PB formation spatially segregates the opposing PHYB signaling actions of PIF5 degradation and stabilization, which provides a PB-enabled counterbalancing mechanism to titrate nucleoplasmic PIF5 and its signaling output[47].

Despite the extensive studies on the dynamic patterns of PBs and the functional roles of PBs in PHYB signaling, the mechanism of PB formation remains ambiguous. One hypothesis is that PBs form via the random LLPS of PHYB in the nucleoplasm[48]. It was recently reported that PHYB could undergo light- and temperature-dependent LLPS into biomolecular condensates in the cytoplasm of mammalian cells[48]. The intramolecular attributes of PHYB required for PB formation in vivo are also essential for PHYB LLPS in this heterologous system, providing evidence that PB formation in plant nuclei may be propelled by the LLPS of active PHYB alone. For example, PHYB's C-terminal module, which is sufficient to form PBs in vivo[35–37], could phase-separate into biomolecular condensates in mammalian cells[48]. The disordered N-terminal extension, which stabilizes the active Pfr form of PHYB[19] and is therefore required for PB formation[36], was also essential for the LLPS of PHYB in mammalian cells[48]. Moreover, the biomolecular condensates of PHYB in mammalian cells could be influenced by temperature change as well[48]. Because the LLPS of PHYB can occur independently of the plant cell, it was proposed that PBs form randomly in the nucleoplasm independently of either cell type or subcellular location[48]. The concept of random LLPS also implies that all PBs within a single nucleus likely possess the same biomolecular composition and biophysical properties.

A significant challenge in applying the framework of LLPS in live cells has been the difficulty in explaining (1) how biomolecules can phase-separate under physiological concentrations that are usually lower than the critical concentration permitting LLPS and (2) how LLPS can generate cellular structures at defined locations. Recent advances in understanding the nucleation of LLPS provide the theoretical basis for localized LLPS in the physiological environment inside live cells. The LLPS of biomolecular condensates requires a nucleation step during the initial stage of phase separation[49–51]. In particular, under physiological conditions where the concentration of biomolecules may be insufficient for global phase separation, sequestering biomolecules at specific slowly diffusing sites serves as a nucleation mechanism that enables localized phase separation[50,51]. Applying the condensation nucleation model to PBs creates an alternative hypothesis that PBs may form nonrandomly at preferred nucleation sites that seed PHYB LLPS in vivo. Our characterization of the temperature-dependent PB dynamics during thermomorphogenesis supports this hypothesis[31]. We identified two types of PBs based on their positioning relative to the nucleolus: (1) nucleolar-associated PBs (Nuo-PBs) that localize at the periphery of the nucleolus and (2) non-nucleolar-associated PBs (nonNuo-PBs) that localize away from the nucleolus[31]. The number of PBs per nucleus is cell/tissue-specific and varies between *Arabidopsis* ecotypes[31]. Intriguingly, increasing the temperature from 12 °C to 27 °C – which slightly destabilizes the active form of PHYB – triggered the disappearance of select PBs[31]. These results suggest that PBs may form at nonrandom subnuclear locations and that individual PBs possess distinct thermosensitivities[31]. However, because PBs were recognized solely based on their nucleolar association, it was uncertain whether

those experiments provided sufficient resolution to distinguish individual PBs.

To determine whether PHYB condensation occurs randomly or nonrandomly in vivo, we established a robust Oligopaints FISH (fluorescence in situ hybridization) method in *Arabidopsis* to label individual PBs. We demonstrate that even in PHYB overexpression lines – where PHYB condensation would be more likely to occur randomly – PBs form at distinct subnuclear locations. Our results reveals a nonrandom PB nucleation model, which provides the framework for the biogenesis of diverse individual PBs at defined places and times in the physiological subnuclear environment.

## Results

### Identification of chromocenter-associated PBs

To assess whether PBs form at nonrandom subnuclear locations, we explored whether we could use chromatin landmarks to distinguish individual PBs. One potential challenge of using chromatin landmarks in *Arabidopsis* is that differentiated cells, particularly the pavement epidermal cells that we usually use to characterize PBs, may undergo variable endoreplication cycles[52]. Because sister chromatids are incompletely aligned in endopolyploid interphase nuclei, labeling most euchromatic regions yields variable numbers of FISH signals either within the same cell type or between different cell types[53]. Therefore, most chromatin regions are unsuitable for use as landmarks. *Arabidopsis* interphase chromosomes are spatially organized into rosette-like chromosome territories[54]. The centromeres and pericentromeric heterochromatin form discrete subnuclear domains called chromocenters (CCs), and euchromatin loops emanate from the CCs[54]. CCs are good landmark candidates because sister-centromeres in endopolyploid interphase nuclei are strictly aligned[53,54]. Labeling the centromeres of the *Arabidopsis* chromosomes ($2n = 10$) using FISH probes against the 178-bp centromeric satellite DNA repeat (*CEN178*) shows a maximum of ten CCs in almost all diploid and endopolyploid differentiated cells[53,54]. The total number of CCs is often less than ten due to frequent association between CCs[54].

We examined the positioning of PBs relative to CCs in the *PBC* (*PHYB-CFP*), a PHYB-CFP overexpression line previously used to characterize PB dynamics in response to light and temperature[31,36]. Because LLPS is concentration-dependent, PHYB condensation would be more likely to occur randomly under a higher PHYB concentration in *PBC* compared with the physiological concentration in Col-0 (wild-type). PBs and CCs were labeled using combined immunostaining and FISH (immunoFISH) to detect PHYB-CFP and *CEN178*, respectively (Supplementary Dataset 1). Because more PBs were present under cool temperatures, the initial experiments were performed using *PBC* seedlings grown in $10\,\mu\text{mol}\,\text{m}^{-2}\,\text{s}^{-1}$ R light at $16\,°\text{C}$[31]. Interestingly, the PBs in the cotyledon pavement cell nuclei could be visually divided into two types based on their positioning relative to CCs: (1) PBs overlapping with or touching a *CEN178* signal, hereafter referred to as chromocenter-associated PBs (CC-PBs), and (2) PBs separated from *CEN178* signals, hereafter referred to as non-chromocenter-associated PBs (nC-PBs) (Fig. 1a). To define the two types of PBs quantitatively, we measured the distance between a PB and the closest CC. Because 2D distance measurements are preferred for comparative analyses of the spatial genome or nuclear organization[55], we quantified the PB-CC distance using maximum-projected confocal images (Fig. 1b). The initial analyses allowed us to establish a cut-off distance of $0.25\,\mu\text{m}$ to distinguish CC-PBs and nC-PBs – i.e., PBs that were $0.25\,\mu\text{m}$ or closer to the nearest *CEN178* signal were defined as CC-PBs, and the rest were categorized as nC-PBs (Fig. 1c). The majority of nuclei ($86.5 \pm 2.0\%$, $n = 521$) contained one to three CC-PBs, only around 5% of nuclei had no CC-PBs (Fig. 1d, e). CC-PBs represented almost half ($47.3 \pm 1.1\%$, $n = 2166$) of the total PBs (Fig. 1f). On average, there were four PBs per nucleus, including two CC-PBs and two nC-PBs (Fig. 1g). Interestingly, for both the CC-PBs and nC-PBs, half localized at the nucleolar periphery and the other half localized away from the nucleolus (Fig. 1g), suggesting that CC association is a feature independent of nucleolar association. However, because there were always several CCs per nucleus, the estimated association frequency of a randomly placed object to all CCs would be too high to be useful for assessing whether the observed PB-CC association was nonrandom.

### Establishment of Oligopaints to label individual CCs

We next asked whether PBs are associated with all or select CCs. Because the CCs contain the centromeric satellite repeats and pericentromeric repetitive transposable elements that are shared among all chromosomes, FISH probes generated using either *CEN178* or bacterial artificial chromosome (BAC) clones of the pericentromeric genomic DNA label all CCs[54,56]. The obvious challenge was to distinguish the individual CCs. To solve this problem, we turned to Oligopaints, which utilizes computationally designed oligos uniquely complementary to the FISH target region[57]. To that end, we first established a robust Oligopaints protocol in *Arabidopsis*. To enhance the FISH efficiency, we incorporated a branched-DNA amplification step that could improve the FISH signal by 32-fold (Fig. 2a)[58]. Oligopaints probes were synthesized to specifically label either the North (top) or South (down) side of the pericentromeric regions of each chromosome (Fig. 2b and Supplementary Dataset 1). The chromosome-specific pericentromeric probes generated FISH signals closely associated with the corresponding CC signals, thereby allowing the recognition of the individual CCs of each chromosome (Fig. 2c). By using the pericentromeric probes of all five chromosomes labeled with a combination of three fluorophores, our Oligopaints method could distinguish all ten CCs within a single nucleus (Fig. 2d, e).

### PBs form nonrandomly near CCs

We then quantified the CC-PBs associated with individual CCs (CC1-5). Interestingly, PBs were associated with all CCs (Fig. 3a–c). There were slightly but significantly more CC2-PBs and CC4-PBs than CC1-PBs, CC3-PBs, and CC5-PBs (Fig. 3a–c). The majority of the cells (over 50%) contained at least one CC2-PB or CC4-PB (Fig. 3a, b), while CC1-PBs, CC3-PBs, or CC5-PBs were found in 43-to-48% of the cells (Fig. 3a, b).

We reasoned that if the association of PBs with a chromatin landmark were random, PBs would be expected to interact with all chromatin landmarks with similar frequencies. Therefore, the fact that PBs showed different association rates with the five CCs suggests that PBs are nonrandomly associated with the CCs. Also, we estimated the random association frequency between a CC and a randomly placed object of a similar size as PBs. The random CC association was estimated to occur in only $15.7 \pm 0.4\%$ of the cell population, which was significantly lower than the experimentally measured PB-CC association frequencies (Fig. 3b). To further assess the nonrandom positioning of CC-PBs, we examined the association of PBs with another chromatin landmark. The *Arabidopsis* genome harbors ten *KNOT ENGAGED ELEMENT*s (*KEE*s) or *INTERACTIVE HETEROCHROMATIC ISLAND*s that are enriched in transposable elements and interact with each other to form several subnuclear structures named the *KNOT*[59,60]. We designed Oligopaints probes specifically labeling *KEE7*, which is located in the South (bottom) arm of chromosome 4 (Fig. 2b). *KEE7* showed no more than two FISH signals in almost all pavement cells, and only $1.8 \pm 1.1\%$ ($n = 112$) of cells showed three signals (Supplementary Fig. 1). In a small fraction of cells ($12.6 \pm 2.9\%$, $n = 112$), *KEE7* showed only one signal, indicative of homologous *KEE7* association (Supplementary Fig. 1). These results indicate that the sister chromatids at the *KEE7* region were aligned in the vast majority of endopolyploid interphase pavement cells. Therefore, the heterochromatic *KEE7* could also be used as a nuclear landmark. Intriguingly, PBs were associated with

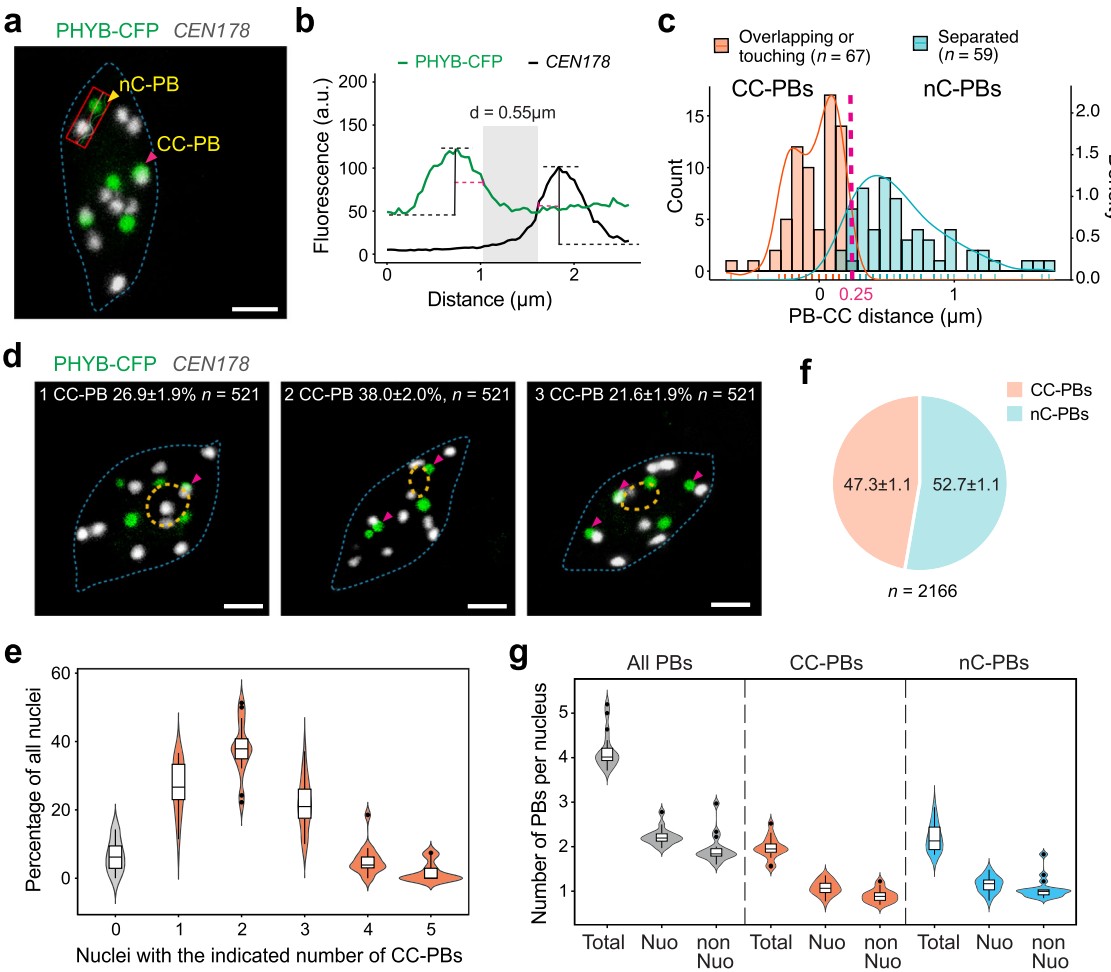

**Fig. 1 | Identification of chromocenter-associated PBs. a** Confocal immunoFISH image showing PBs (green) and CCs (white) in a pavement cell nucleus from the cotyledon of a 4-d-old *PBC* seedling grown under 10 μmol m⁻² s⁻¹ R light at 16 °C. PHYB-CFP PBs were detected via immunofluorescence staining using anti-GFP antibodies, and CCs were labeled via *CEN178* FISH. The magenta arrowhead indicates a CC-PB, whereas the yellow arrowhead indicates a nC-PB. The red box indicates the region of interest (ROI) analyzed in (**b**). This image is a representative image of the 521 analyzed immunoFISH images with similar results. **b** Quantification of the distance between the PB and the nearest CC in the ROI in (**a**). The fluorescence signals of the ROI were quantified to deduce the boundaries of the PHYB-CFP and *CEN178* signals. The boundary of a signal peak was arbitrarily defined as the point with half of the peak fluorescence value (indicated by the dashed magenta line). The PB-CC distance was quantified as the distance between the boundaries of the PHYB-CFP and the *CEN178* signals, corresponding to the width of the shaded area. **c** Distribution plot showing the PB-CC distances of two visually distinct groups of PBs: PBs overlapping with or touching with a *CEN178* signal (orange) and PBs separated from *CEN178* signals (cyan). The vertical dashed magenta line indicates the arbitrary cutoff that distinguishes CC-PBs and nC-PBs. **d** Confocal immunoFISH images showing nuclei containing one to three CC-PBs. CC-PBs are indicated by magenta arrowheads. The numbers indicate the percentage of each type of nucleus and the s.e., *n* indicates the total number of nuclei observed. For (**a** and **d**), the boundaries of the nucleus and nucleolus are traced with dashed blue and orange lines, respectively. Scale bars equal 2 μm. **e** Violin plots showing the percentage of nuclei containing zero to five CC-PBs, *n* = 16 independent FISH experiments. **f** Pie chart showing the proportions of CC-PBs and nC-PBs. **g** Violin plots showing the numbers of all PBs, CC-PBs, and nC-PBs either associated (Nuo) or not associated (nonNuo) with the nucleolus per nucleus, *n* = 16 independent FISH experiments. For (**e**) and (**g**), box plots indicate median (middle line), 25th, 75th percentile (box) and 5th and 95th percentile (whiskers) as well as individually plotted outliers. The source data underlying the PB quantifications in (**b**, **c**, and **e**–**g**) are provided in the Source Data file.

*KEE7* in 29.4 ± 3.4% (*n* = 112) of the pavement cells, which was significantly lower than the association rates between PBs and all CCs (Fig. 3b), lending further support to the conclusion that the association between PBs and CCs was nonrandom.

**Twelve types of spatially distinct PBs**

Next, we characterized the distribution of the CC-PBs and nC-PBs relative to the nucleolus. When considering nucleolar association, the CC-PBs and nC-PBs could be further divided into twelve types, including six types of Nuo-PBs and six types of nonNuo-PBs (Fig. 3c). To identify the major PB types, we examined the average occurrence frequencies of the twelve PB types (Fig. 3c). We used the occurrence frequency of 0.2 PB per nucleus as an arbitrary cutoff to define frequently-occurring and rarely-occurring PBs – i.e., the PB types with

an occurrence frequency above 0.2 PB per nucleus were defined as frequently-occurring PBs (Fig. 3c). The cotyledon pavement cell nuclei from *PBC* seedlings grown at 16 °C harbored nine types of frequently-occurring PBs, including four Nuo-PBs (Nuo-CC2-PB, Nuo-CC3-PB, Nuo-CC4-PB, and Nuo-nC-PB) and five nonNuo-PBs (nonNuo-CC1-PB, nonNuo-CC2-PB, nonNuo-CC3-PB, nonNuo-CC5-PB, and nonNuo-nC-PB) (Fig. 3c). Both the most frequently-occurring Nuo-PBs and nonNuo-PBs were nC-PBs (Fig. 3c), though the Nuo-nC-PB and nonNuo-nC-PB could represent either individual PBs or multiple PB types. There were three rarely-occurring PBs, including Nuo-CC1-PB, Nuo-CC5-PB, and nonNuo-CC4-PB.

The nucleolus-associated CC-PBs comprised mainly Nuo-CC2-PBs and Nuo-CC4-PBs, whereas the non-nucleolus-associated CC-PBs consisted of most nonNuo-CC-PBs except nonNuo-CC4-PBs (Fig. 3c). It was

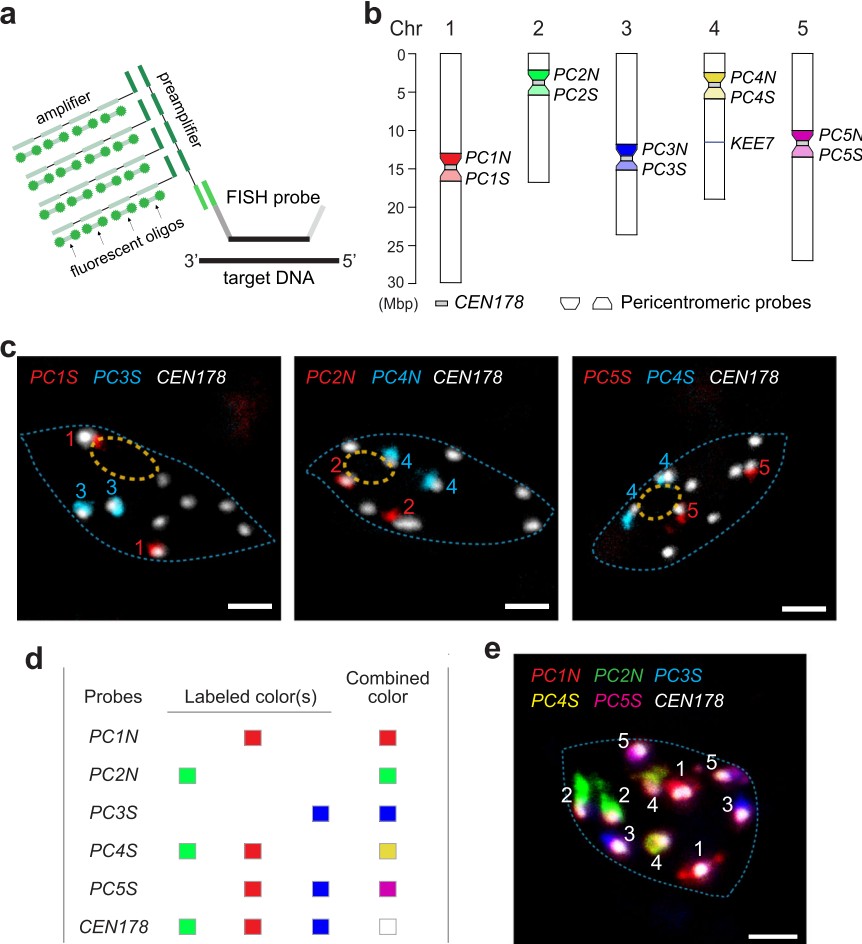

**Fig. 2 | Establishing Oligopaints to label individual chromocenters in *Arabidopsis*.** **a** Schematic illustration of the branched-DNA amplification of Oligopaints signals. Each FISH probe comprises a 36-42-nt sequence (black) complementary to the target DNA, a reverse index (light gray), a reverse transcription primer sequence (dark gray), and a handle sequence (green) containing the binding site for a pre-amplifier. The preamplifier contains four tandem annealing sites for an amplifier, which can recruit four fluorescently labeled oligos. The branched-DNA amplification design could result in a 32-fold multiplexed amplification of the FISH signal. Three sets of the handle, preamplifier, amplifier, and fluorescent oligo sequences were designed to allow the simultaneous labeling of three DNA targets with different fluorophores (see "Methods"). **b** Diagram of the five chromosomes in *Arabidopsis*, depicting the target sites of the *CEN178*, *KEE7*, and pericentromeric FISH probes. **c** Confocal Oligopaints images showing co-labeled *CEN178* CC signals with chromosomal-specific pericentromeric probes in pavement cell nuclei from the cotyledons of 4-d-old *PBC* seedlings grown under 10 μmol m⁻² s⁻¹ R light at 16 °C. These are representative images of at least 60 images for each FISH experiment. The boundaries of the nucleus and nucleolus are traced with dashed blue and orange lines, respectively. Scale bars are equal to 2 μm. **d** Oligopaints probes and their fluorophore colors used for simultaneously labeling ten CCs in a single nucleus in (**e**). **e** Confocal Oligopaints image showing ten CCs labeled with the combinations of three fluorophores depicted in (**d**). The boundary of the nucleus is traced with a dashed blue line. The scale bar equals 2 μm. This is a representative image of at least 30 FISH images with similar results.

previously shown that CC4 and, to a lesser degree, CC2 are preferentially positioned at the nucleolus, correlating with the expression of the 45S rDNA arrays located at these two chromosomal locations[54]. CC1, CC3 and CC5 are primarily localized away from the nucleolus at the nuclear periphery[54]. Consistent with these previous findings, our results showed that CC4 and CC2 were the most frequently nucleolus-associated CCs in pavement cells with nucleolar association frequencies of 76% and 41%, respectively (Fig. 3d). In contrast, CC1, CC3, and CC5 were localized primarily away from the nucleolus, with nucleolar association frequencies less than 25% (Fig. 3d). In general, the nucleolar association rates of CC-PBs correlated with those of the corresponding CCs, explaining the high occurrence of Nuo-CC4-PBs and Nuo-CC2-PBs (Fig. 3c, d). However, the occurrence frequencies of Nuo-CC2-PBs, Nuo-CC3-PBs, and Nuo-CC4-PBs were significantly higher than the occurrence frequencies of the corresponding nucleolus-associated CCs (Fig. 3d), suggesting that PBs may form preferentially at the CCs positioned at the nucleolar periphery.

## The nucleation site dictates PB thermosensitivity

We have previously shown that temperature increases between 12 °C and 27 °C gradually reduce the number of PBs[31]. A reduction in PB number could be due to either PB fusion or PB dissolution. To distinguish these two possibilities, we tested whether all or selective PBs were thermal responsive. We compared the occurrence frequencies of all 12 types of PBs in cotyledon pavement cell nuclei from *PBC* seedlings grown at either 16 °C or 27 °C (Fig. 4a). The three types of rarely-occurring PBs at 16 °C, Nuo-CC1-PB, Nuo-CC5-PB, and nonNuo-CC4-PB, remained rarely-occurring at 27 °C (Fig. 4a). Therefore, we focused on the thermosensitivity of the nine types of frequently-occurring PBs at 16 °C. We defined a PB type as a thermo-sensitive PB if its occurrence frequency at 27 °C decreased at least 30% compared to that at 16 °C. Using this criterion, we identified six types of thermo-sensitive PBs and three types of thermo-insensitive PBs (Fig. 4a). The thermo-sensitive PBs comprise all four frequently occurring nonNuo-CC-PBs as well as Nuo-CC4-PB and Nuo-nC-PB. Interestingly, four thermo-sensitive nonNuo-CC-PBs became rarely-occurring PBs at 27 °C (Fig. 4a), which

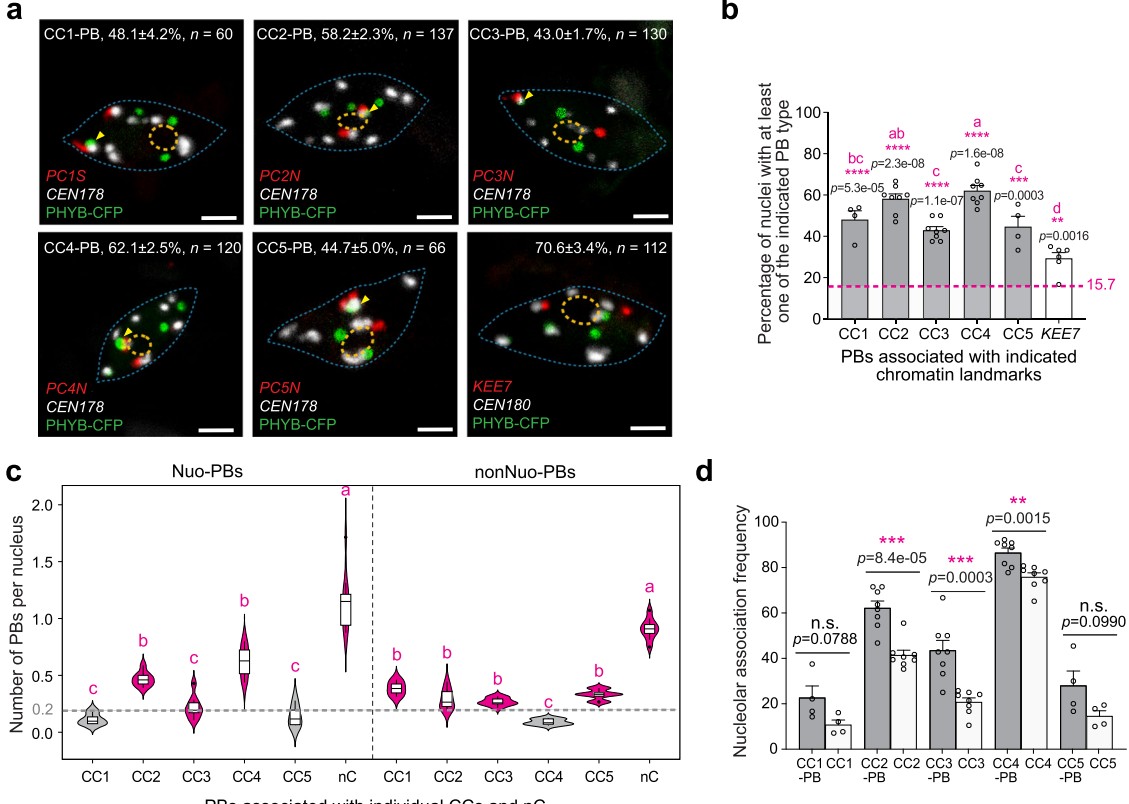

**Fig. 3 | PBs are nonrandomly associated with all chromocenters. a** Confocal immunoFISH images showing that PBs are associated with the CCs of all five chromosomes in cotyledon pavement cell nuclei from 4-d-old *PBC* seedlings grown under 10 μmol m⁻² s⁻¹ R light at 16 °C. Yellow arrowheads indicate CC-PBs. The percentages of nuclei containing at least one indicated PB, including the s.e., are shown; *n* indicates the number of nuclei analyzed. The positioning of PBs relative to *KEE7* was used as a reference control. The boundaries of the nucleus and nucleolus are traced with dashed blue and orange lines, respectively. Scale bars are equal to 2 μm. **b** Quantification of the percentages of cotyledon pavement cell nuclei of the *PBC* seedlings described in (**a**) containing at least one indicated PB type. Error bars represent s.e., *n* = calculated from 4–8 independent immunoFISH replicates. Different letters denote statistically significant differences among the percentages of nuclei containing at least one indicated PB (ANOVA, Tukey's HSD, multiplicity adjusted *p* < 0.05, *n* ≥ 3). The dashed magenta line represents the estimated frequency of association between a CC and a randomly placed object with a similar size as a PB. Asterisks indicate a statistically significant difference between the experimentally determined PB-CC association rate and the estimated random

association frequency based on two-tailed Student's *t*-test (**p* < 0.05, ***p* < 0.01, ****p* < 0.001, *****p* < 0.0001). **c** Quantification of the average occurrence frequencies of individual CC-PBs and the nC-PB in cotyledon pavement cell nuclei from the *PBC* seedlings grown at 16 °C as described in (**a**). Different letters denote statistically significant differences in the occurrence frequency (ANOVA, Tukey's HSD, multiplicity adjusted *p* < 0.05, *n* ≥ 3). Box plots indicate median (middle line), 25th, 75th percentile (box) and 5th and 95th percentile (whiskers) as well as individually plotted outliers. The dashed gray line at 0.2 represents an arbitrary cutoff defining the frequently-occurring and rarely-occurring PBs. The frequently-occurring PBs are highlighted in magenta and the rarely-occurring PBs are labeled in gray.
**d** Comparison of the nucleolar association rates of CC-PBs (gray bars) with those of the corresponding CCs (open bars). Error bars represent s.e., *n* = 4–10 independent immunoFISH replicates. Asterisks indicate a statistically significant difference between the nucleolar association rate of a CC-PB and that of the corresponding CC based on two-tailed Student's *t*-test (**p* < 0.05, ***p* < 0.01, ****p* < 0.001). The source data underlying the PB quantifications in (**b**–**d**) are provided in the Source Data file.

contributed to the decrease of nonNuo-PBs at warmer temperatures[31]. Although the numbers of Nuo-CC4-PBs and Nuo-nC-PBs also decreased substantially in response to the temperate increase, they remained as frequently occurring PBs at 27 °C. We identified three thermo-insensitive PBs; they were Nuo-CC2-PB, Nuo-CC3-PB, and nonNuo-nC-PBs. All three thermo-insensitive PBs were frequently occurring PBs at both low and high temperatures (Fig. 4a.). The previously reported observations based solely on nucleolar association suggested that the thermosensitivity of PBs might correlate negatively with their association with the nucleolus[31]. Our results here demonstrate that this negative correlation may only apply to CC-PBs. When considering all PB, including nC-PBs, there was no correlation between nucleolar association and thermosensitivity. Together, these results suggest that the seeding site determines the thermosensitivity of PBs, which may reflect the temperature-dependent nucleation activity associated with each seeding microenvironment.

Our results support the hypothesis that temperature increases reduce the number of PBs by inducing the disappearance of specific

thermo-sensitive PBs instead of PB fusion. To further support this conclusion, we analyzed nuclei containing only one Nuo-PB or nonNuo-PB at 27 °C. We reasoned that if the reduction of PB number were due to fusion of PBs, there should be much fewer nC-PBs at 27 °C because a fusion of a nC-PB and a CC-PB would yield a CC-PB. Intriguingly, about half of the sole remaining Nuo-PBs and nonNuo-PBs were nC-PBs (Fig. 4b, c), providing further evidence against the PB fusion model. Therefore, we conclude that warm temperatures reduce PB number by inducing the disappearance or attenuating the nucleation of specific thermo-sensitive PBs (Fig. 5).

## Discussion

It has been extensively documented that changes in the light and temperature environment directly regulate the number and size of PBs[12–14,20,31,48]. Although accumulating data suggests that PB formation may be driven by the LLPS of photoactivated PHYB[13,14,48], the mechanism underlying the light- and temperature-dependent regulation of PB formation remains enigmatic. Here we demonstrate that,

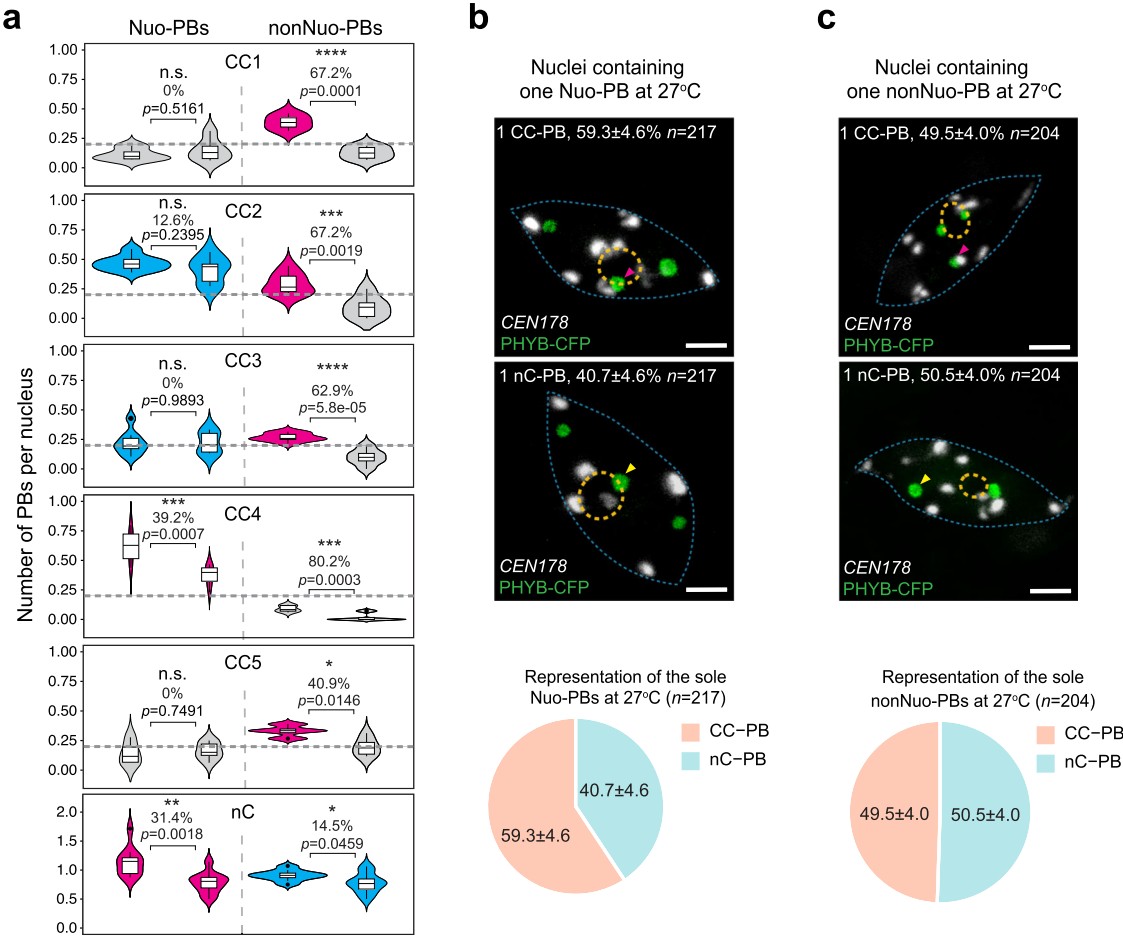

**Fig. 4 | PBs formed at distinct locations possess diverse thermosensitivities.**
**a** Violin plots illustrating the thermosensitivities of the twelve types of PBs in cotyledon pavement cell nuclei from 4-d-old *PBC* seedlings grown under 10 μmol m⁻² s⁻¹ R light at 16 °C or 27 °C. The *p*-value between the data at 16 °C and 27 °C was calculated using Student's two-tailed *t*-test; asterisks indicate statistical significance (**p* < 0.05, ***p* < 0.01, ****p* < 0.001, *****p* < 0.0001), and n.s. indicates not significant. Box plots indicate median (middle line), 25th, 75th percentile (box) and 5th and 95th percentile (whiskers) as well as individually plotted outliers. The dashed gray lines represent the arbitrary cutoff of 0.2 PB per nucleus that defines the frequently-occurring and rarely-occurring PBs. Frequently occurring thermo-sensitive and thermo-insensitive PBs are highlighted in magenta and blue, respectively. Rarely occurring PBs are labeled in gray. **b** Quantification of CC-PBs and nC-PBs in nuclei containing only one Nuo-PB at 27 °C. Upper panel: confocal immunoFISH images showing representative nuclei containing only one Nuo-CC-PB or Nuo-nC-PB.

Bottom panel: pie chart showing the percentage of CC-PBs and nC-PBs as the sole remaining Nuo-PB at 27 °C. The image is a representative image of 217 immunoFISH images with similar results. **c** CC-PBs and nC-PBs in nuclei containing only one nonNuo-PB at 27 °C. Upper panel: confocal immunoFISH images showing representative nuclei containing only one Nuo-CC-PB or Nuo-nC-PB. Bottom panel: pie chart showing the percentage of CC-PBs and nC-PBs as the sole remaining Nuo-PB at 27 °C. The image is a representative image of 204 immunoFISH images with similar results. For (**b** and **c**), PHYB-CFP PBs were detected via immunofluorescence staining using anti-GFP antibodies, and CCs were labeled via *CEN178* FISH. The percentage of the indicated nuclei among the total observed nuclei, including the s.e., is shown in each image. *n* indicates the number of analyzed nuclei. The boundaries of the nucleus and nucleolus are traced by dashed blue and orange lines, respectively. Scale bars are equal to 2 μm. The source data underlying the PB quantifications in (**a**–**c**) are provided in the Source Data file.

unlike the random LLPS of PHYB in heterologous systems[48], PBs form nonrandomly at distinct subnuclear locations in *Arabidopsis* nuclei, even when PHYB-CFP is overexpressed. By using a combination of the nucleolus and individual CCs as nuclear landmarks, we identified twelve types of spatially distinct PBs in the cotyledon pavement cell nuclei, including nine frequently-occurring PBs at 16 °C (Fig. 5). These results reveal a nonrandom PB nucleation mechanism, in which PHYB condensation is seeded at preferred nucleation sites, likely associated with specific chromatin loci. The occurrence frequencies of PBs likely reflect the diverse nucleation activities associated with individual PB seeding environments. The nucleation site also dictates the thermo-sensitivity of PBs. Among the nine frequently-occurring PBs at 16 °C, we identified six thermo-sensitive PBs, including Nuo-nC-PB, Nuo-CC4-PB, nonNuo-CC1-PB, nonNuo-CC2-PB, nonNuo-CC3-PB, and nonNuo-CC5-PB. Unlike the previous observations based solely on nucleolar

association[31], our new results show that the thermosensitivity of PBs does not correlate with nucleolar association. Instead, the thermo-sensitivity of PBs likely reflects the temperature-dependent nucleation activity associated with each seeding microenvironment. We demon-strate that warm temperatures reduce PB number by inducing the disappearance of specific thermo-sensitive PBs (Fig. 5), suggesting that the spatiotemporal control of PB nucleation could represent a pre-viously unknown environmental sensing mechanism.

Accumulating evidence indicates that the LLPS of biomolecular condensates requires a nucleation step during the initial stage of phase separation[49–51]. Under physiological conditions where the concentra-tion of biomolecules may be insufficient for global phase separation, sequestering biomolecules at slowly diffusing sites serves as a nucleation mechanism that seeds localized phase separation[50,51]. The nonrandom PHYB condensation provides evidence supporting the

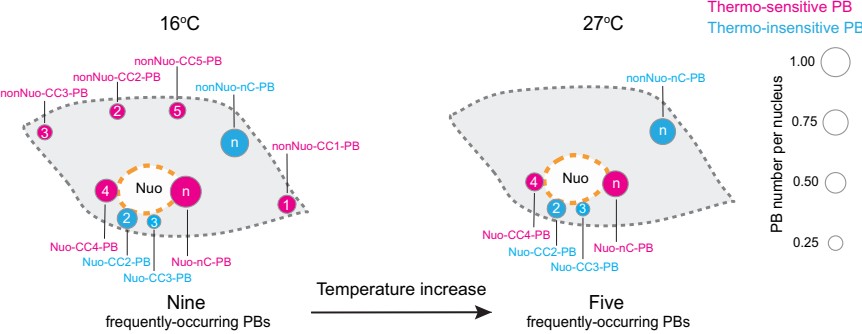

**Fig. 5 | Temperature-dependent PB dynamics.** Schematic illustration of the frequently-occurring PBs in the pavement epidermal cell nuclei at 16 °C and 27 °C. PBs are represented by circles, and the areas of the PB circles are proportional to their occurrence frequencies. The temperature increase from 16 °C to 27 °C induces a dramatic decrease in the occurrence of the six frequently-occurring, thermo-sensitive PBs (labeled in magenta). The temperature increase induces the disappearance of the nonNuo-CC1-PB, nonNuo-CC2-PB, nonNuo-CC3-PB, and nonNuo-CC5-PB, resulting in a dramatic decrease of PB number from nine frequently-occurring PBs at the low temperature to five frequently-occurring PBs at the warmer temperature. The five frequently-occurring PBs at the warm temperature include two remaining thermo-sensitive PBs, Nuo-CC4-PB and Nuo-nC-PB, and three thermo-insensitive PBs (labeled in blue). Dashed gray and orange lines represent the boundaries of the nucleus and the nucleolus, respectively.

localized condensation nucleation model for the LLPS of biomolecular condensates in live cells. Notably, the *PBC* line expresses PHYB-CFP at more than 60-fold the endogenous PHYB in Col-0[47]. Therefore, PHYB condensation would be more likely to occur in *PBC* compared to Col-0. The fact that PHYB condensation still occurred at preferred locations implies either the PHYB-CFP level in *PBC* was still below the critical concentration permitting random global LLPS of PHYB or random PHYB condensation does not occur in vivo. The current immunoFISH method cannot efficiently label endogenous PBs using anti-PHYB antibodies due to low PHYB immunofluorescence signals when combined with FISH. As a result, we could not verify the positioning of PBs in Col-0. However, based on the nonrandom PHYB condensation at an elevated PHYB level in *PBC*, the LLPS theory would predict that PHYB condensation at the low physiological PHYB concentration should also occur nonrandomly at the preferred nucleation sites[47,61].

The concept of condensation nucleation for the phase separation of membraneless organelles in vivo corroborates the previously proposed theoretical framework for nuclear body assembly. Two models have been proposed to explain the self-assembly of nuclear bodies: seeding assembly and stochastic assembly[8]. In fact, both models infer a nucleation mechanism. In the seeding assembly model, nuclear body formation is initiated by the seeding assembly of one or more components at a preferred subnuclear location. A well-studied example of the seeding assembly model is paraspeckles, which are seeded by the long noncoding RNA NEAT1 that recruits its binding proteins to assemble paraspeckles[62]. In the stochastic assembly model, nuclear bodies are assembled via the stochastic interactions of any individual components. The assembly of Cajal bodies fits the stochastic assembly model. However, the formation of Cajal bodies also requires a seeding mechanism[63], as tethering one component to a specific site could nucleate the de novo assembly of a Cajal body[64]. Similarly, tethering PHYB to the nucleolar periphery could also nucleate PHYB condensates and recruit PIF7 to those condensates[65], supporting the idea that PHYB condensation can be nucleated by localized PHYB sequestration. Our results identified at least twelve endogenous PHYB condensation nucleation sites, although we still cannot determine whether the nC-PBs represent individual PBs. The proximity of CC-PBs to the centromeres of all chromosomes suggests that CC-PBs may be seeded at specific pericentromeric chromatin loci. The fact that nC-PBs are positioned away from CCs indicates that nC-PBs are also nucleated nonrandomly at chromosome loci that are spatially separated from the CCs. However, the current data cannot exclude the possibility that nC-PBs are tethered to non-chromatin structures of the nucleus.

Our study unveils that the spatiotemporal control of PHYB condensation nucleation may represent a critical mechanism regulating PB dynamics by warm temperatures in thermomorphogenesis. We showed that changes in ambient temperature alter PB number by controlling the formation of thermo-sensitive PBs (Fig. 4b, c)[31]. These results suggest that individual PB nucleation sites possess diverse temperature-dependent nucleation activities for PHYB condensation. Combined with the observation that the PB number differs within and between cell types[31], the current data suggest that the nonrandom PHYB condensation mechanism may also provide an explanation for cell/tissue-specific PB formation.

The mechanism of PB nucleation remains elusive. The association of PBs with CCs draws a novel link between PB nucleation and specific chromatin loci. In animal cells, the formation of many nuclear bodies, such as the nucleolus, Cajal bodies, and paraspeckles, is associated with the transcription of specific chromatin loci[8,62,63]. Therefore, it is tempting to speculate that the nucleation of PBs also involves transcription at the nucleation site. Supporting this hypothesis, treating seedlings with a Pol II inhibitor abolishes PBs[43]. Five PHYB signaling components have been identified as necessary factors for PHYB condensation. These include PHOTOPERIODIC CONTROL OF HYPOCOTYL 1 (PCH1) and PCH1-LIKE (PCHL) that were identified as PB components by proteomic analysis[46,66–68]. Forward genetic screens for mutants defective in PBs identified HEMERA (HMR)[34], REGULATOR OF CHLOROPLAST BIOGENESIS (RCB)[39], and NUCLEAR CONTROL OF PEP ACTIVITY (NCP)[34,40,69–71]. Further characterization of the functions of PCH1, PCHL, HMR, RCB, and NCP holds great promise for understanding the regulatory mechanism of PB nucleation.

Recent studies suggest that PB formation enables the co-occurrence and counterbalancing of the opposing PHYB actions of PIF5 stabilization in PBs and PIF5 degradation in the surrounding nucleoplasm[47]. It is tempting to speculate that changing the PB number by temperature cues can alter the overall capacity of the PB compartments and rebalance PIF5-mediated environmental responses. The proximity of PBs to CCs may suggest a possible role for PBs in regulating CCs. PHYB plays a prominent role in regulating CC number and compaction[72,73]. Upon light exposure during seedling establishment, PHYB triggers the compaction of pericentromeric heterochromatin into CCs[73]. CC compaction in leaf mesophyll cells is positively regulated by light intensity, and the degree and light response of CC compaction are subject to genetic modification in natural *Arabidopsis* populations during local adaption[72]. Quantitative trait locus (QTL) analysis of natural *Arabidopsis* accessions identified PHYB and HISTONE DEACETYLASE 6 (HDA6) as positive regulators of

light-regulated chromatin compaction[72]. Our results here show that PBs are associated with the CCs of all chromosomes, raising the possibility that PBs are signaling centers that directly regulate chromatin modifications and the spatial organization of CCs.

We could not have identified the distinct PB positioning sites without establishing a new robust Oligopaints FISH method, which enabled the use of individual CCs as landmarks. A similar FISH approach using computationally designed synthetic oligo arrays was reported to paint chromosomal regions in cucumber[74]. To enhance the FISH efficiency, we incorporated a branched-DNA amplification step[58], which could result in a 32-fold amplification of the FISH signal. Moreover, we showed that sister chromatids at one of the interacting heterochromatic *KNOT*s, *KEE7*, were aligned in over 95% of nuclei (Supplementary Fig. 1), suggesting that the *KEE*s can be used as landmarks. The combined Oligopaints design strategy and the improved FISH method enabled us to establish individual CCs and *KEE*s as subnuclear landmarks for investigating spatial nuclear organization in *Arabidopsis*.

Taken together, this study reveals a nonrandom PHYB condensation nucleation mechanism for PB formation, which provides a framework for the biogenesis of spatially distinct PBs with diverse environmental sensitivity within a single nucleus. Our results suggest that the control of PHYB condensation nucleation could represent a previously uncharacterized mechanism in environmental sensing. This study paves the way for investigations of the mechanism of PB nucleation and the functions of individual PBs in light and temperature signaling.

## Methods

### Plant materials and growth conditions
The *PBC* line has been previously described[36]. Seeds were surface-sterilized in 50% bleach with 0.01% Triton X-100 for 10 min and washed four times with sterile ddH$_2$O before being plated on half-strength Murashige and Skoog media with Gamborg's vitamins (MSP06, Caisson Labs, Smithfield, UT), 0.5 mM MES (pH 5.7), and 0.8% (w/v) agar. The seeds were stratified in the dark at 4 °C for five days. For temperature treatments, stratified seeds were placed under continuous 10 µmol m$^{-2}$ s$^{-1}$ R light at 21 °C in an LED chamber (Percival Scientific, Perry, IA) for two days (48 h) to induce seed germination, and then, the seedlings were transferred to either 16 °C or 27 °C under 10 µmol m$^{-2}$ s$^{-1}$ R light and grown for two additional days (48 h). Seedlings were collected 96 h after stratification for all experiments. The fluence rate of light was measured by using an Apogee PS-200 spectroradiometer (Apogee Instruments, Logan, UT).

### Design and synthesis of Oligopaints probe libraries
The Oligopaints probe libraries were designed by following the previously described Oligopaints method[57,75]. A probe library contains a set of oligo probes uniquely complementary to a specific genomic sequence or region. Each probe includes a 36-42-nt genomic sequence flanked by non-genomic sequences, including a universal reverse-transcription primer sequence and a pair of forward and reverse index primers unique to each probe library (Supplementary Fig. 2a). The *CEN178* and pericentromeric probe libraries (*PC1-5N* and *PC1-5S*) consisted of 36-41-nt genomic sequences designed using OligoMiner with the default settings: l 36 -L 41 -g 20 -G 80 -t 42 -T 47 -X "AAAAA;TTTTT;CCCCC;GGGGG"[76]. The *CEN178* probes were designed using the pAL1 centromere sequence as the template[77]. The *KEE7* probe library consisted of 42-nt genomic sequences discovered using OligoArray 2.1[75,78] with the following settings: -l 42 -L 42 -D 50000 -t 78 -T 90 -s 60 -x 55 -p 35 -P 60 -g 45 -m "GGGGGG;CCCCCC;TTTTTT;AAAAAA". The probe sequences of the Oligopaints probe libraries used in this work are listed in Supplementary Dataset 1. The *CEN178*, *PC1-5N*, *PC1-5S*, and *KEE7* probe libraries were synthesized as multiplexed single-stranded oligo pools by GenScript (Piscataway, NJ).

### Oligopaints probe synthesis
Each Oligopaints probe library was amplified from the multiplexed oligo pools purchased from GenScript using PCR with the forward and reverse index primers; the reverse index primer was flanked by a T7 promoter sequence at the 5' end (Supplementary Fig. 2). The resulting double-stranded PCR products were used as the template for in vitro transcription using a HiScribe® T7 Quick High Yield RNA Synthesis Kit (New England Biolabs, Ipswich, MA), followed by reverse transcription (RT) using Maxima H Minus Reverse Transcriptase (EP0753, Thermo Fisher Scientific, Waltham, MA) with either fluorescently labeled RT primers for direct fluorescence labeling of the DNA product or locked nucleic acid (LNA) RT primers for branched-DNA amplification (see below). The in vitro transcription and RT reactions followed the manufacturer's instructions. The *CEN178* probes were synthesized directly as fluorescent probes labeled with ATTO fluorophores; the fluorescently labeled reverse transcription primers used for labeling the *CEN178* probes are listed in Supplementary Table 1. The pericentromeric probe sets and *KEE7* probes were synthesized as LNA probes; the LNA reverse transcription primers used for labeling the pericentromeric probe sets, and *KEE7* probes are listed in Supplementary Table 2. To remove the RNA templates, the RT products were mixed with 4 µl RNaseA (Thermo Fisher Scientific, Waltham, MA) and incubated at 37 °C for 1 h, then at 50 °C for 30 min, and finally at 92 °C for 15 min. The effectiveness of the RT primer incorporation was assessed by running the product on a 10% polyacrylamide gel in a 60 °C water bath for 30 min. The single-stranded DNA probes were purified using a Zymo DNA Clean & Concentrator-100 Kit (Zymo Research, Irvine, CA) following the manufacturer's instructions.

### Nuclear fixation and slide preparation
Seedlings were fixed in 4% (v/v) paraformaldehyde (15710, Electron Microscopy Sciences, Hatfield, PA) with 5% DMSO in PBS for 30 min under vacuum. The fixed seedlings were quenched with three rinses in 50 mM NH$_4$Cl in PBS and then washed twice with PBS. Approximately 100 cotyledons were chopped with a razor blade on a slide in 50 µl lysis buffer (15 mM Tris-HCl pH 7.5, 50 mM EDTA, 0.5 mM spermine-4HCl, 80 mM KCl, 20 mM NaCl, and 0.1% Triton X-100). The resulting suspension containing the released nuclei was transferred into four volumes of nuclei suspension buffer (100 mM Tris-HCl pH 7.5, 50 mM KCl, 2 mM MgCl$_2$, 5% sucrose and 0.05% Tween-20). The nuclei suspension was divided into 30 µl aliquots, spotted on Superfrost® Plus slides (Avantor, Radnor, PA) and air-dried for 4 h at room temperature. The slides were either used immediately or stored at −20 °C.

### ImmunoFISH
The immunoFISH protocol was performed in a 55-µL SecureSeal hybridization chamber (621505, Grace Bio-Labs, Bend, OR). The SecureSeal hybridization chamber was heat-activated and installed over the nuclei sample on a slide following the manufacturer's instructions. The nuclei were washed in MgPBS (1 mM MgCl$_2$ in PBS) and permeabilized for 15 min with MgPBS containing 0.5% Triton X-100. For immunostaining, the slide was washed twice with MgPBS for 5 min and then incubated in a blocking solution containing 2% goat serum, 1.5% bovine serum albumin (BSA), and 0.02% Tween-20 in MgPBS at room temperature for 30 min. Next, the slide was incubated in blocking solution with the primary antibody at 37 °C for 4 h. PHYB-CFP was detected by using either a rabbit anti-GFP polyclonal antibody (A11122, Thermo Fisher Scientific, Waltham, MA) or a chicken anti-GFP polyclonal antibody (A10262, Thermo Fisher Scientific); both were used at 1:100 dilutions. After washing three times with MgPBS containing 0.02% Tween-20 at room temperature, the slide was incubated with secondary antibodies in a blocking solution at 37 °C for 12 h. The

secondary antibodies used in this study included goat anti-rabbit Alexa Fluor 488 antibodies (A32731, Thermo Fisher Scientific), goat anti-rabbit Alexa Fluor 405 antibodies (A31556, Thermo Fisher Scientific), goat anti-chicken Alexa Fluor 488 antibodies (A11039, Thermo Fisher Scientific), and goat anti-chicken Alexa Fluor 405 antibodies (A48260, Thermo Fisher Scientific); all secondary antibodies were used at 1:2000 dilutions. After washing twice with MgPBS containing 0.02% Tween-20, the slide was postfixed with 2% (v/v) paraformaldehyde in MgPBS for 15 min. For Oligopaints, after two 5 min washes with MgPBS, the slide was incubated in 0.1 N HCl for 20 min, rinsed in sterilized water for 5 min, and washed twice in 2× SSCT (0.3 M NaCl, 30 mM sodium citrate and 0.1% Tween-20) for 5 min. The nuclei were incubated for 3 min in 2× SSCT containing 50% formamide (v/v) and 400 µg/mL denatured salmon sperm DNA, then incubated for 20 min at 60 °C in 2× SSCT, and then cooled to room temperature. The nuclei were incubated at 92 °C for 2.5 min for DNA denaturation and then incubated in a hybridization mix, which consisted of 2× SSCT, 50% formamide, 10% dextran sulfate, 40 ng/µL RNase A, and the Oligopaints probes. For *CEN178* FISH, 0.2–0.3 pmol/µl fluorescently labeled *CEN178* probes were used for each experiment. For the pericentromeric and *KEE7* FISH, 0.7–0.8 pmol/µl of LNA probes were used for each experiment. The SecureSeal hybridization chamber was sealed with rubber cement and incubated at 42 °C overnight. The next day, the nuclei were washed three times in 2× SSCT for 10 min. For branched-DNA amplification, the slide was incubated with 50 µl 2× SSCT containing 0.5 pmol of preamplifier at 55 °C for 25 min, washed twice in 2× SSCT for 5 min, incubated in 50 µl 2× SSCT containing 0.5 pmol of amplifier at 55 °C for 25 min, washed twice in 2× SSCT for 5 min, incubated in 50 µl 2× SSCT containing 0.5 pmol fluorescent oligos at 55 °C for 25 min, and washed twice for 10 min in 2× SSCT. On certain slides, the nuclei were counterstained with 500 ng/mL 4′,6-diamidino-2-phenylindole (DAPI) in 2× SSCT at room temperature for 30 min, then washed three times with 2× SSCT for 10 min. Finally, the SecureSeal hybridization chamber was removed, and the slide was mounted in ProLong Gold Antifade Mountant (P10144, Thermo Fisher Scientific, Waltham, MA).

### Confocal microscopy

Three-dimensional (3D) image stacks of individual pavement epidermal nuclei were imaged using a Zeiss LSM800 confocal microscope with a 100×/1.4 oil-immersion objective lens. Alexa Fluor 405 was excited with a 405 nm laser and monitored using a 400–570 nm bandpass emission. ATTO488 and Alexa Fluor 488 were excited with a 488 nm laser and detected using a 400–560 nm bandpass emission. ATTO555 was excited with a 561 nm laser and observed using a 560–630 nm bandpass emission. ATTO647 was excited with a 640 nm laser and monitored using 645–700 nm bandpass emission. 3D images were taken as Z-stacks with 0.7 µm intervals through each nucleus. Maximum projections of the image stacks were obtained using Zeiss ZEN 2.3 software (Carl Zeiss AG, Jena, Germany) for subsequent analysis or exported as TIFF files and processed using Adobe Photoshop CC (Adobe, San Jose, CA).

### Estimating the random CC-PB association frequency

The random association frequency between a CC and an object of a similar size as a PB was estimated as described previously, with minor modifications[79]. The average dimensions of PBs, *CEN178* signals, and the nuclear area ($S_{Nu}$) were measured using ZEN 2.3 software. Precisely, the radius was calculated as the distance from the signal's peak to its boundary, which was defined at the point where the fluorescence intensity was half of the peak value. The $r_{PB}$ and $r_{CC}$ are the average radii of PBs and *CEN178* signals within each nucleus, respectively. A cutoff distance of 0.25 µm was used as the threshold for the edge-to-edge association between a PB and a CC. The random CC-PB association frequency was calculated using the following formula:

$$\text{Random CC} - \text{PB association frequency(\%)} = 100 \times [\pi(r_{PB} + 0.25 + r_{CC})^2 / S_{Nu}] \times 2$$

### Reporting summary

Further information on research design is available in the Nature Portfolio Reporting Summary linked to this article.

## Data availability

The *Arabidopsis* lines and the reagents used in the current study are available from the corresponding author upon request. Source data are provided with this paper.

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

## Acknowledgements

We are grateful to Chao-ting Wu and Huy Quoc Nguyen (Harvard Medical School) for their assistance with Oligopaints. We thank the High-Performance Computing Center at the University of California Riverside for providing assistance with the bioinformatics tools used in the study. We thank Elise Pasoreck for valuable suggestions and comments on the manuscript. This work was supported by National Institute of General Medical Sciences grant R01GM087388 and National Science Foundation grant MCB-2141560 to M.C.

## Author contributions

J.D., K.K. and M.C. conceived the original research plan; M.C. supervised the experiments; J.D., K.K. and M.C. performed the experiments; J.D. and M.C. analyzed the data and wrote the article.

## Competing interests

The authors declare no competing interests.
