## [Peer Review File · Nature Communications]

Distinguishing individual photobodies using Oligopaints reveals nonrandom thermo-sensitive and -insensitive phytochrome B condensation at distinct subnuclear locationsReviewer #1 (Remarks to the Author):

The manuscript by Du et al. described the quantitative measurement of photobodies (PBs) comprised of Pfr-phyB and their relative sub-nuclear localization with chromocenters (CC) and nucleolus. The authors also examined the temperature sensitivity of the photobodies in proximity to CC or nucleolus and observed some differences in the dissolution of photobodies nucleated at different subnuclear locations under a higher temperature.

1. If this reviewer understands correctly, the PBC lines express phyB driven by the constitutive promoter 35S. Whether the phyB-CFP level is comparable or much higher than the endogenous phyB level should be discussed. The complementation of this construct of the phyB mutant only indicates the phyB-CFP is biologically functional. However, it doesn't mean the cytological behaviors of photobodies composed of phyB-CFP reflect the endogenous phyB-photobodies, especially LLPS can be highly concentration dependent.
2. The results were primarily descriptive and technically oriented. The appearance of photobodies near CC or nucleolus is intriguing. Still, their biological relevance to light or temperature responses needed to be clarified. Based on the current results, it is difficult to assess whether the unique localization of these photobodies is at all important.
3. Perhaps, as the authors stated in the "Discussion," how key genes needed for full PB formation should be examined to really understand the underlying mechanisms of the nucleation behaviors of PBs near CC and nucleolus and the possible contribution of the CC-PB/nc-PB in the spatial organization of chromatin structure and CC formation.
4. In this manuscript, 2D measurement was conducted and used to arbitrarily determine the 0.25 um cutoff distance for CC-PBs and nC-PBs. Yet, some nC-PBs still have a distance shorter than 0.25 um (Fig. 1C). It's unclear whether the CC and nC classification was ambiguous and whether it's more helpful to derive the cutoff from 3D images instead of the 2D ones.
5. It's unclear why a random CC-PB association is expected to be in 15.7% of the cell population (Fig. 3b). If this is the case, then PBs should not be associated with KEE in 29.4% of the cell population. Wouldn't this imply that PBs also tend to associate with KEE non-randomly?
6. CC2 and CC4 have been reported to be associated with nucleolus; therefore, the observations in this study were not new. If PBs preferentially associate with CC2 and CC4, then their association with nucleolus is a natural consequence driven by CC2 and CC4. Results in this study cannot differentiate the "nucleation" of PBs that occur at CC2/CC4 before or after their association with nucleolus.
7. The authors tried to conclude that CC-PBs have different thermal stabilities, determined by their association with nucleolus or not, but the difference was trivial. The authors also claimed higher temperature induces the dissolution of thermally unstable PBs. It needs to be clarified how the reduced assembly rate cannot explain this phenomenon.
8. The Oligopaint FISH works well and could interest a broader audience. The author may consider communicating the work to a more technically oriented journal.

Reviewer #2 (Remarks to the Author):

The manuscript by Du et al demonstrate that phytochrome B condensation is not random and might differentiate thermostability of photobodies. Authors developed fascinating technique, where "painting" with different colors of different chromocenters was possible. It is very elegant technique that allows correlation of localization between multiple factors. Even though I truly like the technique and appreciate the discovery of the paper I do have few comments, that are crucial for me to further understand the story behind the discoveries.

How can efficiency affect the FISH labeling of all CCs? Do authors have control on that? In theory, if efficiency of oligopaint FISH is different between the probes it might lead to different perception of the colors thus lead to different conclusion on number of CCs.

Authors made a test for the temperature; however it is well established that changes of PBs happen because of temperature or light or both. incorporating the light aspect would be important for testing the system having another factor influencing the system.

Authors showed different – non-random association of PBs with five CCs, what is the biological relevance of that and what implication it would have if one CC would be modified?

Authors submitted two parallel stories. It seems that multiple parts of the manuscript is identical. Authors have to check the text carefully to avoid plagiarism of the other paper. For example: Line 76 – Joanna Chory, identical text like in the Kim et al paper recently submitted
Line 48 – avoid using semicolon. More elegant is to start new sentence.
Line 166-168 – please correct the sentence for clarity.

Response to Reviewers

We thank both reviewers for their thorough reviews of the manuscript and for their constructive comments and suggestions. We made substantial changes to the manuscript. We modified Fig 3. to define nine frequently-occurring PBs and three rarely-occurring PBs at 16°C. This allowed us to better define thermo-sensitive PBs in Fig. 4. We also revised the model (Fig. 5) to illustrate the occurrence frequencies of all frequently-occurring PBs at both low and high temperatures and highlight the effect of warm temperatures in reducing the number of thermo-sensitive PBs.

Reviewer #1

The manuscript by Du et al. described the quantitative measurement of photobodies (PBs) comprised of Pfr-phyB and their relative sub-nuclear localization with chromocenters (CC) and nucleolus. The authors also examined the temperature sensitivity of the photobodies in proximity to CC or nucleolus and observed some differences in the dissolution of photobodies nucleated at different subnuclear locations under a higher temperature.

1. If this reviewer understands correctly, the PBC lines express phyB driven by the constitutive promoter 35S. Whether the phyB-CFP level is comparable or much higher than the endogenous phyB level should be discussed. The complementation of this construct of the phyB mutant only indicates the phyB-CFP is biologically functional. However, it doesn't mean the cytological behaviors of photobodies composed of phyB-CFP reflect the endogenous phyB-photobodies, especially LLPS can be highly concentration dependent.

Response:

We thank the reviewer for the comment. We added the following in the second paragraph in the Discussion section: “It is important to note that the *PBC* line expresses PHYB-CFP at more than 60-fold the endogenous PHYB in Col-0⁵⁰. The fact that PHYB condensation still occurred at preferred locations at this elevated PHYB level implies that the PHYB-CFP level in *PBC* was still below the critical concentration permitting random global LLPS. Therefore, PHYB condensation at the physiological PHYB concentration must also require a nucleation step^{50,64}.”

2. The results were primarily descriptive and technically oriented. The appearance of photobodies near CC or nucleolus is intriguing. Still, their biological relevance to light or temperature responses needed to be clarified. Based on the current results, it is difficult to assess whether the unique localization of these photobodies is at all important.

Response:

The findings of nonrandom nucleation of PBs, in our opinion, represent a groundbreaking discovery in the plant photobiology and thermomorphogenesis fields as well as the general field of nuclear organization. Our findings provide evidence validating the recently proposed, general condensation nucleation model for the LLPS of biomolecular condensates in living cells [Bracha et al. (2019) *Cell* 175:1467; Shimobayashi et al. (2021) *Nature* 599:503]. The current model of PB formation proposed by the recent study by Chen et al. [*Mol Cell* 82:3015 (2022)] posits that

“photo-activated phyB undergoes LLPS independently of either cell type or subcellular location”. This *Mol Cell* paper was considered “paradigm-shifting” to understanding light and temperature signaling in plants [Lee and Huq (2022) *Mol Cell* 82:2916]. Our results here argue against the random PHYB LLPS model for PB formation. The current study combined with our previous work [Hahm et al. (2020) *Nat Commun* 11:1660] demonstrates that PB formation occurs at preferred subnuclear locations and is cell/tissue specific. We agree with the reviewer that the establishment of the Oligopaint FISH protocol in *Arabidopsis* represents a major technical breakthrough. In fact, we spent more than four years setting up the Oligopaint FISH method in *Arabidopsis*. However, our goal in developing the method was to determine the mechanism of PB formation. Our findings of the nonrandom seeding of individual PBs and their diverse thermal sensitivities open a new avenue for investigating both the mechanism and significance of PB formation in light and temperature sensing. We agree with the reviewer that we still do not fully understand the significance of the unique localization of individual PBs. But, we have to say that it is this study that raises this novel question. Without this study, we would not have even noticed the importance of the specific localizations of individual PBs. Even if someone could have asked the question, there would not have been a tool to explicitly define and recognize the individual PBs without this study.

3. Perhaps, as the authors stated in the “Discussion,” how key genes needed for full PB formation should be examined to really understand the underlying mechanisms of the nucleation behaviors of PBs near CC and nucleolus and the possible contribution of the CC-PB/nc-PB in the spatial organization of chromatin structure and CC formation.

Response:

We agree with the reviewer that this study only uncovers the nucleation requirement and defines the localizations of PB formation. We would need future molecular genetics and biochemical studies to understand the underlying mechanism of PHYB condensation nucleation.

4. In this manuscript, 2D measurement was conducted and used to arbitrarily determine the 0.25 μm cutoff distance for CC-PBs and nC-PBs. Yet, some nC-PBs still have a distance shorter than 0.25 μm (Fig. 1C). It's unclear whether the CC and nC classification was ambiguous and whether it's more helpful to derive the cutoff from 3D images instead of the 2D ones.

Response:

It has been well-established in the nuclear organization field that 2D distance measurements are preferred for comparative analyses of spatial genome/subnuclear organization [Finn et al. (2017) *Methods* 123:47]. Instead of defining CC-PBs based on visual observations of separation between PB and CC, which is subject to variations, we established the definition of CC-PBs and nC-PBs based on quantitative distance measurements between the PHYB-CFP PB and the closest *CEN178*-FISH-labeled CC. The cutoff of 0.25 μm between CC-PBs and nC-PBs was determined experimentally (Fig. 1c). As shown in Fig. 1c, a small fraction of visually-defined “Separated” PBs would be categorized as CC-PBs using the quantitative method.

5. *It's unclear why a random CC-PB association is expected to be in 15.7% of the cell population (Fig. 3b). If this is the case, then PBs should not be associated with KEE in 29.4% of the cell population. Wouldn't this imply that PBs also tend to associate with KEE non-randomly?*

Response:

We estimated the random CC-PB association frequency based on a previously reported method for estimating a random association between Cajal bodies and the chromosome territories [Wang et al. (2016) *Nat Commun* 7:10966]. The detailed method was described in the Methods section “Estimating the random CC-PB association frequency”. As stated in the manuscript, our data indicate that the PB-*KEE7* association frequency was above the random association frequency. It is conceivable that any chromatin loci linked to the PB seeding site should exhibit a nonrandom PB-association frequency, the association frequency would reflect the distance between the chromatin locus and the PB seeding site.

6. *CC2 and CC4 have been reported to be associated with nucleolus; therefore, the observations in this study were not new. If PBs preferentially associate with CC2 and CC4, then their association with nucleolus is a natural consequence driven by CC2 and CC4. Results in this study cannot differentiate the “nucleation” of PBs that occur at CC2/CC4 before or after their association with nucleolus.*

Response:

We agree that the nucleolar association of CC-PBs depends on the association of the CCs to the nucleolus (Fig. 3d). As stated in the manuscript: “CC4 and, to a lesser degree, CC2, are preferentially positioned at the nucleolus, correlating with the expression of the 45S rDNA arrays located in these two chromosomes, whereas the other CCs are primarily localized to the nuclear periphery (away from the nucleolus)⁵⁴. Consistent with these previous findings, our results showed that CC4 and CC2 were the most frequently nucleolus-associated CCs in pavement cells under our experimental conditions, displaying nucleolar association in 70% and 40% of cells, respectively (Fig. 3d).” Our results also indicate that PBs can form independently of the nucleolar-association of CC4 and CC2, because PBs form at both nucleolar-associated and non-nucleolar-associated CC2s (Figs. 3c and 4a).

7. *The authors tried to conclude that CC-PBs have different thermal stabilities, determined by their association with nucleolus or not, but the difference was trivial. The authors also claimed higher temperature induces the dissolution of thermally unstable PBs. It needs to be clarified how the reduced assembly rate cannot explain this phenomenon.*

Response:

We define thermosensitive PBs as those whose occurrence frequency – measured by the number of a particular type of PBs per nucleus – drops more than 30% at 27°C compared to 16°C. For example, the occurrence of non-nucleolar-associated CC1-, CC2-, and CC3- PBs dropped more than 60% from 16°C to 27°C. A 60% difference in a total dynamic range of 100% is enormously

significant, not “trivial” at all. Our results show that the thermosensitive CC1-, CC2-, CC3-, and CC5-PBs almost disappeared at 27°C – we revised our model (Fig. 5) to highlight this point.

The reviewer raised an interesting point about the difference between PB nucleation and assembly. The LLPS nucleation model proposes that under physiological conditions where the concentration of biomolecules may be insufficient for global phase separation, sequestering biomolecules at specific slowly diffusing sites serves as a nucleation mechanism that enables localized phase separation [Bracha et al. (2019) *Cell* 175:1467; Shimobayashi et al. (2021) *Nature* 599:503]. So, the PB nucleation mechanism presumably involves higher assembly rates for PHYB condensation. In that sense, PHYB condensation nucleation includes the assembly rate or may not be mechanistically different from the assembly rate.

8. The Oligopaint FISH works well and could interest a broader audience. The author may consider communicating the work to a more technically oriented journal.

Response: We thank the reviewer for the positive comments on the Oligopaint FISH method. Please see our responses to Question #2.

Reviewer #2:

The manuscript by Du et al demonstrate that phytohrome B condensation is not random and might differentiate thermostability of photobodies. Authors developed fascinating technique, where "painting" with different colors of different chromocenters was possible. It is very elegant technique that allows correlation of localization between multiple factors. Even though I truly like the technique and appreciate the discovery of the paper I do have few comments, that are crucial for me to further understand the story behind the discoveries.

How can efficiency affect the FISH labeling of all CCs? Do authors have control on that? In theory, if efficiency of oligopaint FISH is different between the probes it might lead to different perception of the colors thus lead to different conclusion on number of CCs.

Response:

As the reviewer indicated, the robustness of the FISH method is critical for the analysis. One advantage of the Oligopaint FISH method is to use computationally-designed oligos with similar T_m values, therefore ensuring similar binding affinities to different target sequences. Our robust Oligopaint FISH method made it possible to consistently label individual CCs as nuclear landmarks.

Authors made a test for the temperature; however it is well established that changes of PBs happen because of temperature or light or both. incorporating the light aspect would be important for testing the system having another factor influencing the system.

Response:

As the reviewer pointed out, changes in both light and temperature alter PHYB conformation and the pattern of PBs. The main difference is that altering light quantity and quality elicits a more dramatic change in both PB number and size, which makes it more complicated to track individual PBs. For example, transitions to dimmer light or shade lead to the localization of PHYB-FPs to tens of small PBs [Hahm et al. (2020) *Nat Commun* 11:1660]. To tackle the difficult question of PB formation, we decided to use the simplest model first, which is the formation of a few large PBs under strong light and the reduction of PBs in warmer temperatures [Hahm et al. (2020) *Nat Commun* 11:1660]. Now, with the knowledge of the PB nucleation mechanism, we can ask next what happens to PB nucleation in dim light or shade conditions and whether the small PBs represent new seeding sites or a result of random disassembly of the large PBs.

Authors showed different – non-random association of PBs with five CCs, what is the biological relevance of that and what implication it would have if one CC would be modified?

Response:

This is an interesting question. Obviously, we still do not know the answer to this question. But, we can imagine that if the number of PBs determines the total capacity of the PB compartments in a nucleus, a change in the number of PBs could alter the PB/nucleoplasmic partitioning of PHYB and other PB constituents and therefore their signaling functions in both PBs and the surrounding nucleoplasm.

Authors submitted two parallel stories. It seems that multiple parts of the manuscript is identical. Authors have to check the text carefully to avoid plagiarism of the other paper. For example: Line 76 – Joanna Chory, identical text like in the Kim et al paper recently submitted

Response: We thank the reviewer for the advice. We have revised the text to eliminate identical sentences between the two manuscripts.

Line 48 – avoid using semicolon. More elegant is to start new sentence.

Response: We revised the sentence as the reviewer suggested.

Line 166-168 – please correct the sentence for clarity.

Response: We revised the sentence.

Reviewer #2 (Remarks to the Author):

I see that authors addressed all points suggested by my previous revision. I do not agree with one comment regarding efficiency and robustness of FISH. Even if primers are optimized and set to be as similar as possible, still the efficiency can be affected by hybridization efficiency and penetration.

Otherwise, I do not have further comments.

Reviewer #3 (Remarks to the Author):

I have not reviewed the initial version of this manuscript but have been invited to comment on the revised version and the authors response to the reviewers of the first version of the manuscript.

Formation of photobodies is an intriguing feature of the plant photoreceptor phytochrome B (phyB). It has been observed the first time more than two decades ago, but their molecular nature and to a large extent also their physiological function remained elusive. Work in the last few years provided support for the idea that photobodies are membrane-less subnuclear compartments formed by liquid-liquid phase separation. In the work presented in this manuscript, the authors established Oligopaint FISH to label individual chromocentres (CC) and used this method to investigate a link between chromocentres and the nucleation of photobodies. The authors show that photobodies form preferentially at twelve nucleation sites associated with chromocentres or the nucleolus. They also find that a subset of the photobodies is temperature-sensitive and disappears at elevated temperature.

Overall, this is an interesting manuscript that introduces a new view on how photobodies form. As such, it certainly has the potential to have a significant impact on the field. However, I agree with reviewer #1 and #2 that there are still a number of open questions that should be addressed and I have some additional comments as detailed below.

Major comments:

1. In the last sentence of the abstract, the authors write "... and unveils that the spatiotemporal control of PHYB condensation nucleation represents a critical mechanism in plant light and temperature signaling." and the last sentence of the introduction is "Our study reveals that the spatiotemporal control of the nucleation of PHYB condensation represents a critical mechanism in plant light and temperature signaling."  What is the experimental evidence for that? I would agree that there is evidence (based on previous publications by numerous labs) that photobodies play a role in light signalling. However, the authors claim that non-random PB nucleation is crucial. This conclusion would require mutants that are unable to form non-random PBs – I don't see how else one could support this conclusion.
2. This comment is in the same direction as a comment by one of the reviewers. I wonder how the distance between two points in 3D can be measured based on an image in 2D? I.e. points that are close to each other with regard to the x- and y-axis might be far away on the z-axis or do I misunderstand that. I am not expert on microscopy and image processing, i.e. I don't want to say that the method is not appropriate, but it would be great if the authors could clarify this point (other non-experts might have the same question).
3. On line 315ff the authors write "Together, these results suggest that the thermosensitivity of PBs is determined by the seeding site and likely reflects the temperature-dependent nucleation activity associated with each seeding microenvironment."  This would be a phyB-independent effect, if I understand this sentence correctly. Or is the hypothesis that thermal reversion of phyB is different for photobodies nucleated at different sites? Wouldn't one expect that all photobodies are temperature-sensitive given that only phyB Pfr assembles into photobodies and the stability of phyB Pfr is reduced at elevated temperature?
4. The authors propose that photobodies form nonrandomly at distinct subnuclear locations (chromocentres, nucleolus). How about the hundreds of smaller photobodies that form in a bit weaker light? Would they also form nonrandomly at predefined nucleation sites?
5. The authors write in the discussion line 362f "Therefore, PHYB condensation at the physiological

PHYB concentration must also require a nucleation step ...". Is this really a valid conclusion? What if there were a factor that has to be present at equimolar level as phyB and that allows phyB to form photobodies independently of predefined nucleation sites? Can it be ruled out that overexpressed phyB artificially assembles with nucleation sites and drags all phyB into the few very large photobodies? I am a bit hesitant to believe such a very strong statement ('must also require') without experimental evidence from a line expressing phyB at WT level.

6. In lines 394f, the authors speculate about a link between the nucleation sites (chromocentres) and transcription. What genes would that be? Do genes related to phytochrome signalling locate near chromocentres? What nucleation sites are used by other nuclear bodies (e.g. CRY2 nuclear bodies)? Do the authors have any idea why phyB photobodies should be associated with chromocentres, while other nuclear bodies are not? It would have been interesting to test the localisation of other nuclear bodies and if they indeed do not overlap with phyB photobodies.

7. Line 437ff: The authors write "Our results uncover the control of PHYB condensation nucleation as a critical mechanism in light and temperature signaling."  In my opinion, this statement is not supported by experimental evidence. The authors show data suggesting that phyB photobodies nucleate at specific sites, but this does not mean that nucleation at these specific sites is essential or of physiological relevance. This would require mutants in which phyB PB nucleation occurs randomly.

8. The frequency of many PBs is around or even less than 0.25, i.e. only every fourth cell contains the respective PBs. Would that mean that cells containing the respective PBs are different from cells not containing it with regard to phyB-mediated responses? What is the author's view on this?

9. Regarding response to question 1 by reviewer #1: Mimicking lower levels of phyB Pfr by using lower light intensities results in many small photobodies. Thus, a lower concentration of phyB Pfr would lead to a different pattern of photobodies, which would suggest that the pattern of photobodies observed for the 60x overexpression line may not correspond with the photobody pattern in a line expressing WT levels of phyB. Can the authors comment on that?

10. Regarding response to question 2 by reviewer #1: I fully agree with reviewer #1. Although the authors highlight the novelty of their approach, they do not answer reviewer #1's question whether the unique localisation of photobodies is important at all.

11. Regarding response to question 1 (regarding efficiency of FISH labelling) by reviewer #2: The authors do not really answer the reviewer's question. Do they have a control to ensure equal efficiency of labelling?

12. Regarding response to question 3 ("Authors showed different – non-random association of PBs with five CCs, what is the biological relevance of that and what implication it would have if one CC would be modified?) by reviewer #2: The authors response is "Obviously, we still do not know the answer to this question."  I fully agree with this response ... but in the manuscript, the authors have statements like "... unveils that the spatiotemporal control of PHYB condensation nucleation represents a critical mechanism in plant light and temperature signaling." or similar, which imply that they have answered the question.

Minor comments:

13. I think throughout the manuscript the role of phyB as temperature sensor is exaggerated. The activity of phyB is affected by temperature and therefore temperature has effect on a subset of physiological responses under specific conditions. However, saying "... temperature changes are perceived ... by PHYB..." (line 86ff) implies that phyB is THE temperature sensor in plants which is not true.

14. Lines 95ff: The authors argue that CRY2 NBs and phyB PBs are different because there is only partial overlap of CRY2 NBs and phyB PBs. However, a partial overlap could also be interpreted in the way that CRY2 NBs and phyB PBs are not entirely independent. Any comments on this view?

Response to Reviewers

Reviewer #2

I see that authors addressed all points suggested by my previous revision. I do not agree with one comment regarding efficiency and robustness of FISH. Even if primers are optimized and set to be as similar as possible, still the efficiency can be affected by hybridization efficiency and penetration. Otherwise, I do not have further comments.

Response: Maybe we did not quite understand the question previously. In our immunoFISH experiments, we mostly used at least two FISH probes, one against the *CEN178* centromeric repeats and the other against a specific pericentromeric region. The *CEN178* probe serves as technical control of the FISH experiment because *CEN178* FISH should always work.

Reviewer #3

Formation of photobodies is an intriguing feature of the plant photoreceptor phytochrome B (phyB). It has been observed the first time more than two decades ago, but their molecular nature and to a large extent also their physiological function remained elusive. Work in the last few years provided support for the idea that photobodies are membrane-less subnuclear compartments formed by liquid-liquid phase separation. In the work presented in this manuscript, the authors established Oligopaint FISH to label individual chromocentres (CC) and used this method to investigate a link between chromocentres and the nucleation of photobodies. The authors show that photobodies form preferentially at twelve nucleation sites associated with chromocentres or the nucleolus. They also find that a subset of the photobodies is temperature-sensitive and disappears at elevated temperature.

Overall, this is an interesting manuscript that introduces a new view on how photobodies form. As such, it certainly has the potential to have a significant impact on the field. However, I agree with reviewer #1 and #2 that there are still a number of open questions that should be addressed and I have some additional comments as detailed below.

We thank Reviewer #3 for the thorough review of our manuscript and for the constructive and thoughtful comments.

Major comments:

1. In the last sentence of the abstract, the authors write "... and unveils that the spatiotemporal control of PHYB condensation nucleation represents a critical mechanism in plant light and temperature signaling." and the last sentence of the introduction is "Our study reveals that the spatiotemporal control of the nucleation of PHYB condensation represents a critical mechanism in plant light and temperature signaling."  What is the experimental evidence for that? I would agree that there is evidence (based on previous publications by numerous labs) that photobodies play a role in light signalling. However, the authors claim that non-random PB nucleation is crucial. This conclusion would require mutants that are unable to form non-random PBs – I don't see how else one could support this conclusion.

Response: We thank the reviewer for the comments. The current model of PHYB LLPS, which was proposed by Chen et al. [*Mol Cell* 82:3015 (2022)] based on expressing PHYB in mammalian cells, posits that “photo-activated phyB undergoes LLPS independently of either cell type or subcellular location”. Unlike this model, our conclusion is that LLPS of PHYB occurs nonrandomly *in vivo*. This is true even at 60-fold the endogenous PHYB level. Therefore, either 60-fold endogenous PHYB is still under the critical concentration that permits random LLPS or random LLPS of PHYB does not occur *in vivo*. If PB formation requires nucleation at a preferred location, the nucleation activity at individual seeding sites would be critical. This is supported by the temperature-sensitive PBs as they disappear at warm temperatures. The main conclusion of our study is that PB formation occurs nonrandomly at preferred sites – this means PBs either form at those sites or do not form at all – mutants that cannot form non-random PBs would be those that do not form PBs.

We revised the sentence: “unveils that the spatiotemporal control of PHYB condensation nucleation may represent a critical mechanism in plant light and temperature signaling.”

2. This comment is in the same direction as a comment by one of the reviewers. I wonder how the distance between two points in 3D can be measured based on an image in 2D? I.e. points that are close to each other with regard to the x- and y-axis might be far away on the z-axis or do I misunderstand that. I am not expert on microscopy and image processing, i.e. I don't want to say that the method is not appropriate, but it would be great if the authors could clarify this point (other non-experts might have the same question).

Response: It is well established in the nuclear organization field that 2D measurements are preferred for comparative analyses of spatial genome/subnuclear organization [Finn et al. (2017) *Methods* 123:47]. There are three basic points to help understand the process. First, the goal of the analysis is not to accurately measure the distance between two objects but rather to assess their association frequency. Therefore, by projecting a 3D image stack to a 2D image, the distance between the two objects in the 2D image should reflect their relationship in the 3D space. Second, the scenario of two non-associated objects aligning on the z-axis is a rare event. You can imagine the two objects being the Sun and the Moon – a solar eclipse is rare. Third, Because FISH is a single-molecule technique and biological events are mostly stochastic, it would be critical to qualify the behavior of single molecules in a population. In our results, most association results, such as PB-CC associations, were assessed using over 100 data points (some over 500 data points) – i.e., over 100 3D nuclei images that were projected to 2D images for analysis.

3. On line 315ff the authors write "Together, these results suggest that the thermosensitivity of PBs is determined by the seeding site and likely reflects the temperature-dependent nucleation activity associated with each seeding microenvironment."  This would be a phyB-independent effect, if I understand this sentence correctly. Or is the hypothesis that thermal reversion of phyB is different for

photobodies nucleated at different sites? Wouldn't one expect that all photobodies are temperature-sensitive given that only phyB Pfr assembles into photobodies and the stability of phyB Pfr is reduced at elevated temperature?

Response: The conclusion that the seeding site determines the thermosensitivity of PBs does not mean it is PHYB-independent. But, different from the previous model, PB formation would depend on both PHYB conformation and the nucleation site. The nucleation model posits that the seeding sites may possess high PHYB-binding affinity to increase the local PHYB concentration. Then, the PHYB binding affinity can vary between the nucleation sites. These intrinsic variations among the individual seeding sites could discriminate their abilities to recruit the less-stable active PHYB at warm temperatures. On the other hand, the strong-seeding sites' binding affinity to PHYB could remain high even with the less-stable active PHYB at warm temperatures.

4. The authors propose that photobodies form nonrandomly at distinct subnuclear locations (chromocentres, nucleolus). How about the hundreds of smaller photobodies that form in a bit weaker light? Would they also form nonrandomly at predefined nucleation sites?

Response: This is a great question. Under dim light and shade conditions, PHYB-FP localizes to tens of small PBs. We had intentionally tried to avoid the small PBs by only looking at the large PBs under strong light conditions, because the large PBs are a more straightforward starting point – which is already complex enough to take us years to get to this point. The new results encouraged us to rethink the model for small PBs. It would be interesting to test whether the small PBs are also formed non-randomly.

5. The authors write in the discussion line 362f "Therefore, PHYB condensation at the physiological PHYB concentration must also require a nucleation step ...". Is this really a valid conclusion? What if there were a factor that has to be present at equimolar level as phyB and that allows phyB to form photobodies independently of predefined nucleation sites? Can it be ruled out that overexpressed phyB artificially assembles with nucleation sites and drags all phyB into the few very large photobodies? I am a bit hesitant to believe such a very strong statement ('must also require') without experimental evidence from a line expressing phyB at WT level.

Response: We thank the reviewer for the comments. We revised the sentence: “PHYB condensation at the physiological PHYB concentration likely also requires a nucleation step^{50,64}.”

6. In lines 394f, the authors speculate about a link between the nucleation sites (chromocentres) and transcription. What genes would that be? Do genes related to phytochrome signalling locate near chromocentres? What nucleation sites are used by other nuclear bodies (e.g. CRY2 nuclear bodies)? Do the authors have any idea why phyB photobodies should be associated with chromocentres, while other nuclear bodies are not? It would have been interesting to test the localisation of other nuclear bodies and if they indeed do not overlap with phyB photobodies.

Response: We do not know the nucleation sites at this point. There are too many possibilities. We have yet to look at other nuclear bodies. Colocalization studies using BY-2 and *Arabidopsis* protoplasts suggest that the PHYB-containing PBs and the CRY2-containing nuclear bodies represent distinct subnuclear domains [Mas et al (2000) *Nature* 408:207; Kim et al (2023) *Nat Commun* 14:1708].

7. Line 437ff: The authors write "Our results uncover the control of PHYB condensation nucleation as a critical mechanism in light and temperature signaling."  In my opinion, this statement is not supported by experimental evidence. The authors show data suggesting that phyB photobodies nucleate at specific sites, but this does not mean that nucleation at these specific sites is essential or of physiological relevance. This would require mutants in which phyB PB nucleation occurs randomly.

Response: We revised the sentence: "Our results unveil that the control of PHYB condensation nucleation may represent a previously uncharacterized mechanism in PHYB signaling."

8. The frequency of many PBs is around or even less than 0.25, i.e. only every fourth cell contains the respective PBs. Would that mean that cells containing the respective PBs are different from cells not containing it with regard to phyB-mediated responses? What is the author's view on this?

Response: Interesting question! As shown in Fig. 4b and c, nuclei containing only one Nuo-PB could have different PBs. This would be an exciting question to address in the future.

9. Regarding response to question 1 by reviewer #1: Mimicking lower levels of phyB Pfr by using lower light intensities results in many small photobodies. Thus, a lower concentration of phyB Pfr would lead to a different pattern of photobodies, which would suggest that the pattern of photobodies observed for the 60x overexpression line may not correspond with the photobody pattern in a line expressing WT levels of phyB. Can the authors comment on that?

Response: Reducing light intensity does not mimic lower levels of active phyB per se, as it would also reduce the percentage of time that individual PHYB molecules stay in the Pfr form. Our recent preprint (<https://www.biorxiv.org/content/10.1101/2023.11.12.566724v3>) described three PHYB overexpression lines expressing various PHYB levels, *PBC* (60x endogenous PHYB), *gPBC-29* (40x endogenous PHYB), and *gPBC-25* (7x endogenous PHYB). There was no noticeable difference in PB number in these lines. We also showed PBs of endogenous PHYB via immunostaining using antibody PHYB antibodies (Fig. 2d); these PBs were similar to those in the transgenic lines.

10. Regarding response to question 2 by reviewer #1: I fully agree with reviewer #1. Although the authors highlight the novelty of their approach, they do not answer reviewer #1's question whether the unique localisation of photobodies is important at all.

Response: We thought we had answered reviewer #1's question. The importance of nonrandom seeding of PBs is that 1) this is essential for PB formation **in vivo** and 2) it allows the biogenesis of distinct PBs with diverse thermosensitivity.

11. *Regarding response to question 1 (regarding efficiency of FISH labelling) by reviewer #2: The authors do not really answer the reviewer's question. Do they have a control to ensure equal efficiency of labelling?*

Response: Please see our response to Reviewer #2.

12. *Regarding response to question 3 ("Authors showed different – non-random association of PBs with five CCs, what is the biological relevance of that and what implication it would have if one CC would be modified?) by reviewer #2: The authors response is "Obviously, we still do not know the answer to this question."  I fully agree with this response ... but in the manuscript, the authors have statements like "... unveils that the spatiotemporal control of PHYB condensation nucleation represents a critical mechanism in plant light and temperature signaling." or similar, which imply that they have answered the question.*

Response: We modified this statement throughout the manuscript.

Minor comments:

13. *I think throughout the manuscript the role of phyB as temperature sensor is exaggerated. The activity of phyB is affected by temperature and therefore temperature has effect on a subset of physiological responses under specific conditions. However, saying "... temperature changes are perceived ... by PHYB..." (line 86ff) implies that phyB is THE temperature sensor in plants which is not true.*

Response: We did not suggest that PHYB was the only thermal sensor. We basically discussed PHYB conformational changes and PB formation based on the context of the current literature. If PB is driven by PHYB condensation alone, a direct effect of temperature on PBs would be temperature-dependent PHYB thermal reversion.

14. *Lines 95ff: The authors argue that CRY2 NBs and phyB PBs are different because there is only partial overlap of CRY2 NBs and phyB PBs. However, a partial overlap could also be interpreted in the way that CRY2 NBs and phyB PBs are not entirely independent. Any comments on this view?*

Response: The current data suggest these two nuclear bodies represent distinct subnuclear domains [Mas et al (2000) *Nature* 408:207; Kim et al (2023) *Nat Commun* 14:1708]. Suppose two subnuclear domains are partially overlapping in all nuclei or occasionally overlapping in some nuclei. In our opinion, they should be regarded as distinct subnuclear compartments.

Reviewer #3 (Remarks to the Author):

I thank the authors for their responses.

The authors argue that non-random formation of PBs in lines overexpressing phyB 60-fold proves that non-random formation of PBs does not occur *in vivo*. However, this is not particularly convincing in my opinion. Can they exclude that PBs aggregate with CCs only due to artificially high levels? They mention that immunolocalisation of phyB demonstrates that phyB PBs in wildtype background (no overexpression of phyB) show a similar localisation as in the line 60x overexpressing phyB. Both reviewer #1 and I were concerned about potential artefacts by overexpression of phyB. Using wildtype seedlings for immunolocalisation and oligopaint/labeling of CCs could confirm that phyB associates with CCs even when present at wildtype levels. Is there any reason why the authors did not do this experiment which would strongly support their claims? Furthermore, even if phyB associates with CCs when present at wildtype levels, this does not necessarily mean that phyB function depends on this association. Such a statement would require a line with random phyB PB formation. If such a line is impaired in proper phyB responses, the conclusion that non-random PBs formation is essential for phyB function would be valid.

I also wonder if existence of mutants with random localisation of PBs can be excluded? What is the argument for that? How does phyB associate with CCs/nucleation sites? Can the authors exclude that there is a factor that acts as nucleation site for PBs and that mediates association of PBs with CCs? A hypothetical mutant could be affected in association with CCs but still be functional with regard to nucleating PBs. The existence of the hundreds of small PBs clearly demonstrates that PBs can form independently of CCs.

Response to Reviewer #3

Reviewer #3 (Remarks to the Author):

I thank the authors for their responses.

The authors argue that non-random formation of PBs in lines overexpressing phyB 60-fold proves that non-random formation of PBs does not occur in vivo. However, this is not particularly convincing in my opinion. Can they exclude that PBs aggregate with CCs only due to artificially high levels? They mention that immunolocalisation of phyB demonstrates that phyB PBs in wildtype background (no overexpression of phyB) show a similar localisation as in the line 60x overexpressing phyB. Both reviewer #1 and I were concerned about potential artefacts by overexpression of phyB. Using wildtype seedlings for immunolocalisation and oligopaint/labelling of CCs could confirm that phyB associates with CCs even when present at wildtype levels. Is there any reason why the authors did not do this experiment which would strongly support their claims? Furthermore, even if phyB associates with CCs when present at wildtype levels, this does not necessarily mean that phyB function depends on this association. Such a statement would require a line with random phyB PB formation. If such a line is impaired in proper phyB responses, the conclusion that non-random PBs formation is essential for phyB function would be valid.

Response:

We appreciate the reviewer's comments and have taken them into consideration. According to the LLPS theory, LLPS is concentration-dependent. In the PHYB overexpression lines, PHYB is more likely to phase separate randomly. We have revised the title, abstract, and text to emphasize this point and also clarified that the current immunoFISH method cannot efficiently label the PBs in Col-0 using anti-PHYB antibodies. This study, while addressing the important question of whether PBs form randomly or nonrandomly in the nucleoplasm, also opens up intriguing avenues for future investigations into the functional significance of nonrandom PB formation.

We also included the following in the Discussion section:

“Notably, the *PBC* line expresses PHYB-CFP at more than 60-fold the endogenous PHYB in Col-0⁴⁷. Therefore, PHYB condensation would be more likely to occur in *PBC* compared to Col-0. The fact that PHYB condensation still occurred at preferred locations implies that the PHYB-CFP level in *PBC* was still below the critical concentration that permits random global LLPS of PHYB or random PHYB condensation does not occur at all *in vivo*. The current immunoFISH method cannot efficiently label endogenous PBs using anti-PHYB antibodies due to low PHYB immunofluorescence signals when combined with FISH. As a result, we could not verify the positioning of PBs in Col-0. However, based on the nonrandom PHYB condensation at an elevated PHYB level in *PBC*, the LLPS theory would predict that PHYB condensation at the low physiological PHYB concentration should also occur nonrandomly at the preferred nucleation sites^{47,61}.”

I also wonder if existence of mutants with random localisation of PBs can be excluded? What is the argument for that? How does phyB associate with CCs/nucleation sites? Can the authors exclude that there is a factor that acts as nucleation site for PBs and that mediates association of PBs with CCs? A hypothetical mutant could be affected in association with CCs but still be functional with regard to nucleating PBs. The existence of the hundreds of small PBs clearly demonstrates that PBs can form independently of CCs.

Response:

This study marks the first exploration into the nonrandom localization of PBs. Prior to our research, this question had not been posed, and the tools to characterize PB positioning were not available. Our study, while revealing the nonrandom seeding of PBs, also highlights the gaps in our understanding, particularly in the seeding mechanism. We cannot exclude any mechanism for recruiting PHYB to the seeding sites. The potential isolation of a seeding mutant in the future would be a significant advancement. Importantly, we have demonstrated that 50% of PBs are not associated with CC, emphasizing that CC association is not a requirement for all PBs.